



# Observation and modelling of atmospheric OH and HO₂ radicals at a subtropical rural site and implications for secondary pollutants

Zhouxing Zou[1#], Tianshu Chen[1#], Qianjie Chen[1], Weihang Sun[1], Shichun Han[1], Zhuoyue Ren[2], Xinyi Li[2], Wei Song[2], Aoqi Ge[2], Qi Wang[2], Xiao Tian[2], Chenglei Pei[3], Xinming Wang[2], Yanli Zhang[2], and Tao Wang[1]

[#] These authors contributed equally to this work

[1] Department of Civil and Environmental Engineering, The Hong Kong Polytechnic University, Hong Kong, China

[2] Guangzhou Institute of Geochemistry, Chinese Academy of Sciences, Guangzhou, China

[3] Guangdong Province Guangzhou Ecological Environment Monitoring Center Station, Guangzhou 510030, China.

Correspondence to: Tao Wang (tao.wang@polyu.edu.hk)

**Abstract**

HO_X radicals (OH and HO₂) are crucial oxidants that determine atmospheric oxidation capacity and the production of secondary pollutants; however, their sources and sinks remain incompletely understood in certain forest and maritime environments. This study measured HO₂ and OH concentrations using a chemical ionisation mass spectrometer at a subtropical rural site in southern China from 12 November to 19 December 2022. The average peak concentrations were $3.50 \pm 2.47 \times 10^6$ cm$^{-3}$ for OH and $1.34 \pm 0.93 \times 10^8$ cm$^{-3}$ for HO₂. Calculations based on an observation-constrained chemical model revealed an overestimation of HO₂ and OH concentrations during warm periods of the field study. These inaccuracies resulted in overestimations of production rates in the model simulation by up to 98% for ozone and 341% for nitric acid. Our study highlights the need for further improving understanding of the sources/sinks of OH and HO₂.

**1. Introduction**



The $HO_X$ family, comprising hydroxyl (OH) and peroxy radicals ($HO_2$), plays a
pivotal role in the Earth's atmosphere by driving photochemical processes that influence
the air composition and chemistry. OH radicals are primarily produced by the
photolysis of ozone ($O_3$), nitrous acid (HONO), and ozonolysis of alkenes. They initiate
the oxidation of CO and most volatile organic compounds (VOCs), producing $HO_2$ and
other peroxyl radicals ($RO_2$, where R represents an alkyl group). $HO_2$ is also generated
from the photolysis of oxygenated VOCs (OVOCs) and by reactions between OVOCs
and OH. In the presence of NO, $RO_2$ radicals are converted to $HO_2$ and then to OH
radicals buffering OH concentrations and maintaining atmospheric oxidation capacity.
(Stone et al., 2012). These interactions are crucial in the formation of photochemical
smog and secondary organic aerosol (SOA), which generate $NO_2$, $O_3$ and highly
oxygenated molecules. $HO_X$ radicals are removed through reactions of OH with
inorganic trace gases, self-reactions among radicals, peroxyacetyl nitrate (PAN)
formation, and the heterogeneous uptake by aerosols, subsequently contributing to
atmospheric acidification and aerosol formation by the production of $H_2SO_4$ and $HNO_3$.
See Text S1, Figure S1 and Table S1 for detailed processes and chemical reactions.
The accuracy of model-predicted OH is a crucial indicator for assessing our
understanding of the atmosphere processes (Heard and Pilling, 2003). There is a
longstanding debate regarding the discrepancies between simulated and observed
radical concentrations under low $NO_x$ condition which remains a significant issue
(Hofzumahaus et al., 2009; Stone et al., 2012; Zou et al., 2023). Previous studies have
shown that models generally predict OH levels well in polluted conditions (NO > 1
ppb), but notable overestimation were observed under low NO and aged conditions,
such as coastal areas (Kanaya et al., 2007; Zou et al., 2023), marine boundary layers
(Berresheim et al., 2002; Carslaw et al., 1999), and the Arctic region. Missing OH sinks
were proposed as the primary reason for the overestimation (Lou et al., 2010; Yang et
al., 2016; Hansen et al., 2014 Thames et al., 2020). Underestimation of OH
concentrations were also observed in high biogenic VOCs (BVOCs) and low NO (<1
ppb) conditions (Hofzumahaus et al., 2009; Lelieveld et al., 2008; Tan et al., 2001;



Whalley et al., 2011). After considering a new OH regeneration mechanism (Wennberg
et al., 2018; Novelli et al., 2020) and a measurement interference (Feiner et al., 2016;
Hens et al., 2014; Mao et al., 2012; Novelli et al., 2014; Woodward-Massey et al., 2020),
daytime OH concentration could be reasonably reproduced by the model in the high
BVOC conditions, with some unresolved underestimation in the evening (Jeong et al.,
2022; Lew et al., 2020; Tan et al., 2019).

7       $HO_2$ concentrations were consistently underpredicted in the polluted urban sites

(Ma et al., 2019; Yang et al., 2021; Ma et al., 2022) and at a rural site (Tan et al., 2017),
with no clear trends in relatively clean regions. Some studies reported good agreement
between measurement and model prediction (Feiner et al., 2016; Lew et al., 2020),
whereas others indicated model overprediction (Bottorff et al., 2023; Griffith et al.,
2013) and underprediction (Kim et al., 2013; Mallik et al., 2018) . These discrepancies
may be attributed to several factors, including: measurement interference caused by
$RO_2$ recycling in environments rich in BVOCs or aromatics (Fuchs et al., 2011),
uncertainties associated with heterogeneous uptake in box models (Yang et al., 2022),
unmeasured sinks and the outflow of reservoir species like PAN (Griffith et al., 2013).
Despite these advances, it remains difficult to pin down the exact causes of the model-
measurement discrepancies in some of the previous studies.
In the present study, we measured concentrations of OH and $HO_2$ using a
quadrupole chemical ionization mass spectrometer (PolyU-CIMS) from November to
December 2022 at a subtropical rural site of southern China. We test model's capability
in reproducing the radical concentrations and elucidate factors contributing to
discrepancy under varying temperature, VOCs, and $NO_X$ conditions. The Methodology
section describes the measurement site, the principle and the configuration of PolyU-
CIMS, and the setup of chemical box model. The Results and Discussion section details
our findings, providing a comprehensive analysis of the radical concentrations and
exploring the discrepancies between observed data and model predictions. By
employing an observation-constrained box model, we analyzed the radical budgets and





investigated potential causes for these discrepancies. The study concludes with a
discussion of the implications of these findings.
**2. Methodology**
**2.1 Measurement Site**

5       The field campaign was conducted at the Conghua Liangkou Air Monitoring

Station (23°44'47"N, 113°47'06"E, 200m, above sea level) from November 12 to
December 19, 2022 (Figure 1). The site is located at the northern part of the Pearl River
Delta (PRD), approximately 80 kilometers from the densely populated areas of the PRD
and nestled within the Liuxi River National Forest Park (an evergreen broad-leaf forest).
The site is situated just north of the G105 national highway and around 0.5 kilometer
east from Liangkou town. Even though it is close to the road, the traffic was generally
limited during the observation period due to the coronavirus disease pandemic (COVID
19). The site is subjected to the BVOC emission, predominantly isoprene, from the
surrounding forest when the daytime temperature is exceeding 20°C, and NO emissions
from the nearby national highway, particularly during periods of low wind speeds. The
measurements comprised trace gases including $O_3$, NO, $NO_2$, CO, HONO, VOCs,
OVOCs, meteorological data such as relative humidity (RH), temperature, and
photolysis frequencies of HONO, $NO_2$, $O_3$, $H_2O_2$, and HCHO. Details about the
instruments are shown in Table S2.



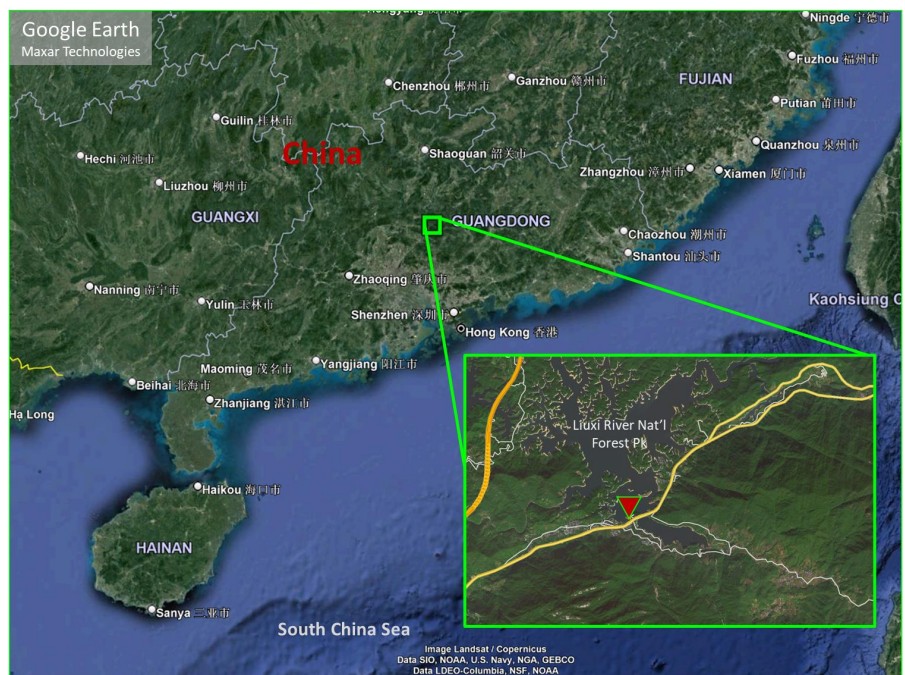

**Figure 1** Geographic location of measurement site (Liangkou Air Monitoring station 23°44'47"N, 113°47'06"E, 200 m a.s.l. labelled by the red inverted triangle) in Conghua, Guangdong Province, South China. The map is sourced from © Google Earth and © Amap.

**2.2 Radical measurement principle**

OH and HO$_2$ radicals were measured using the Hong Kong Polytechnic University quadrupole Chemical Ionization Mass Spectrometer (PolyU-CIMS), which was used in a previous study (Zou et al., 2023) for OH measurement. The use of CIMS for OH measurement was pioneered by Eisele and Tanner, (1991), with subsequent enhancements in measurement accuracy (Eisele and Tanner, 1993; Tanner et al., 1997; Tanner and Eisele, 1995) and adoption of inlets for simultaneous measurements of HO$_2$ and RO$_2$ (Edwards et al., 2003; Sjostedt et al., 2007), H$_2$SO$_4$ (Mauldin III et al., 2004), and OH reactivity (Muller et al., 2018).

Figure 2 illustrates the measurement principle of the PolyU-CIMS used in this campaign. Briefly, the ambient OH radicals are converted to H$_2$SO$_4$ in the sample inlet system by reacting with SO$_2$ (R21 in the reaction Table S1) which is then transformed to HSO$_4^-$ ion clusters in the ionization chamber by the reactions with a reagent gas in

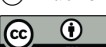



sheath flow (HNO$_3$, R24 to 27), and ultimately dissociated (R29) for detection by the
mass spectrometer system at m/z 97 (S$_{97SO2}$ in Figure 2). To mitigate interference and
noise, scavenger gases (C$_3$F$_6$ in this study) were introduced to scavenge the ambient
OH, creating a background signal (R23, S$_{97ScaSO2}$ in Figure 2). The ambient OH radicals
signal (S$_{OH}$) is then determined by the subtracting S$_{97ScaSO2}$ from S$_{97SO2}$. The OH
concentration is calculated using the following equation:
$$[OH] = \frac{1}{C_{OH}} \times \frac{S_{OH}}{S_{62}} \quad \text{(E1)}$$
Where C$_{OH}$ represents the calibration factors of OH, and S$_{62}$ is the signal corresponding
to the reagent ion (NO$_3^-$). The detailed calibration procedure for OH is outlined in
previous studies (Kürten et al., 2012; Zou et al., 2023).

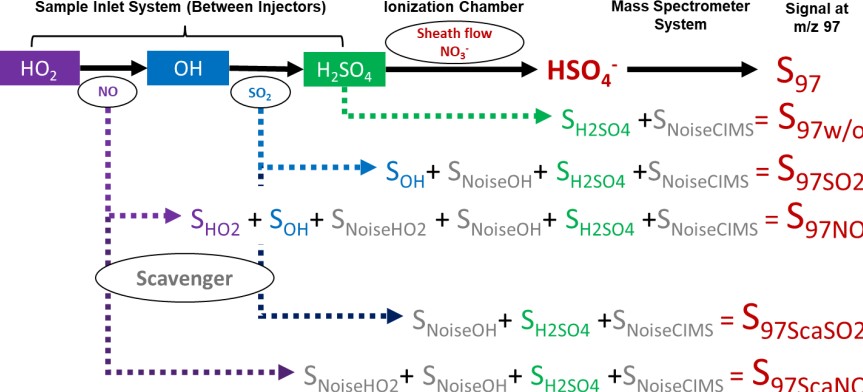

**Figure 2** Flow chart depicting the relationship between measurement species and signal intensity at
m/z 97 (S$_{97}$) with various gas injection. The color-filled grids labeled the ambient species, while
oval shape label the species injected into the sample flow. Signal intensities with different gas
additions to the sample flow are represented by S$_{97w/o}$, S$_{97SO2}$, S$_{97NO}$, S$_{97ScaSO2}$, and S$_{97ScaNO}$. The
signals corresponding to ambient OH, HO$_2$, H$_2$SO$_4$ and noise from OH measurement, HO$_2$
measurement and the CIMS denoted as S$_{OH}$, S$_{HO2}$, S$_{H2SO4}$, S$_{NoiseOH}$, S$_{NoiseHO2}$ and S$_{NoiseCIMS}$,
respectively.
To measure ambient HO$_2$, NO is injected into the sample flow, converting HO$_2$ to
OH (R11). This converted OH then follows the same reaction pathway (R21, R24 to
R27, and R29) and is measured at m/z 97 (S$_{97NO}$ in Figure 2). Similar to the OH
measurement, the background signal for HO$_2$ (S$_{97ScaNO}$ in Figure 2) is determined by
introducing the scavenger gas. The corresponding signal for ambient HO$_2$ (S$_{HO2}$ in



Figure 2) is determined by subtracting $S_{97ScaNO}$ and $S_{OH}$ from $S_{97NO}$. The $HO_2$
concentration is calculated using a similar equation to E1, by replacing $S_{OH}$, and $C_{OH}$ to
$S_{HO2}$ and $C_{HO2}$, respectively (E2).

$$[HO_2] = \frac{1}{C_{HO2}} \times \frac{S_{HO2}}{S_{62}} \ (E2)$$

The procedure for determination of $HO_2$ calibration factor, $C_{HO2}$, are akin to that for
$C_{OH}$. The calibration tube generates equal amounts of radicals (R30 in SI, [OH]/[$HO_2$]
= 1), allowing for simultaneous calibration of $HO_2$ and OH with and without NO
addition to the sample flow.
Compared to its configuration in the previous campaign (Zou et al., 2023), the
PolyU-CIMS has been upgraded for simultaneous $HO_2$ measurements. See Figure S2,
Text S2 on modification for $HO_2$ measurement and interference. Apart from the
modifications, the PolyU-CIMS's setting, and configurations remained same as those
in the previous campaign. (Table S3). With the updated configuration, the PolyU-CIMS
achieved the simultaneous measurement for the three gases.
The calibration factor, detection limit and accuracy were $1.09 \times 10^{-8}$ cm$^{-3}$, $3 \times 10^5$
cm$^{-3}$, and 46% for OH; $6.01 \times 10^{-9}$ cm$^{-3}$, $2 \times 10^6$ cm$^{-3}$, and 44% for $HO_2$; and $1.09 \times 10^{-8}$
$^8$ cm$^{-3}$, $1 \times 10^5$ cm$^{-3}$, and 40% for $H_2SO_4$, respectively (Table S3).
**2.3 Box Model**
$HO_X$ concentrations in this study were simulated using the Framework for 0-D
Atmospheric Modelling (F0AM, Wolfe et al., 2016) with the Master Chemical
Mechanism (MCM) v3.3.1 (http://mcm.leeds.ac.uk/MCM), which encompasses over
6700 species and 17000 reactions. MCM v3.3.1 features a near-explicit chemical
mechanism, including isoprene degradation and OH regeneration mechanisms. This
mechanism has been previously employed for investigating $HO_X$ chemistry and
conducting budget analyses (Slater et al., 2020; Tan et al., 2018; Zou et al., 2023). The
gas-phase chlorine chemistry described by Xu et al., (2015) and Wang et al. (2019)
were included in the model (Chen et al., 2022).
In the baseline scenario, the observation data were aggregated into one-hour
intervals to provide input for the model, initially constraining it without incorporating



observed OH and $HO_2$ data. For the assessment of ozone formation rates, the model
was adjusted to include constraints based on the actual measured concentrations of OH
and $HO_2$. Observed VOCs were categorized into anthropogenic origin (AVOCs),
including species from petroleum gas and industrial solvent evaporation (alkenes,
alkenes, benzene, and TEXs - toluene, ethylbenzene, and xylenes), and OVOCs
comprising aldehydes, ketones, and acids. The sole BVOC measured in this study was
isoprene. Methacrolein (MACR), a derivative of isoprene, is distinctively classified
among the biogenically sourced OVOCs for further discussion. Physical processes like
deposition and entrainment in the model were represented by a first-order physical loss
with a 24-hour lifetime for all species (Chen et al., 2022; Wolfe et al., 2016; Zou et al.,
2023). The model also included the heterogeneous uptake of $HO_2$ by aerosols,
represented as a pseudo-first order loss (Jacob, 2000):

$$\frac{d[HO_2]}{dt} = -k_{HO_2}[HO_2] \text{ (E3)}$$

$$k_{HO_2} = \frac{V_{HO_2} \times S_a \times \gamma_{HO_2}}{4} \text{ (E4)}$$

$$v_{HO_2} = \sqrt{\frac{8RT}{\pi \times M_{HO_2}}} \text{ (E5)}$$

Here, $k_{HO_2}$ represents the first-order loss rate coefficient of $HO_2$ by aerosol uptake,
determined by the effective $HO_2$ uptake coefficient $\gamma_{HO2}$ (0.1, Guo et al., 2019), the
mean molecular velocity of $HO_2$ ($v_{HO_2}$), the aerosol surface area concentration ($S_a$)
measured by the Scanning Mobility Particle Sizing (SMPS); and the molecular mass of
$HO_2$ ($M_{HO_2}$= 17 g/mol). As aerosol and aqueous phase chemistry were not included in
the model, it was assumed that the heterogeneous $HO_2$ loss would not lead to further
reactions (Guo et al., 2019). For each day, a three-day spin-up was performed with
constant inputs to establish stable model chemistry and reduce the uncertainty of
unconstrained species.

25        In addition to simulating ambient concentrations, the model was also utilized to

estimate an inlet interference which is the OH radicals recycled from the reaction of
ambient $HO_2$ and NO in the inlet. The model's reaction time was set to 47 ms (matching
the reaction time in the sample inlet), photolysis frequencies were set to zero, and the



injection gases ($SO_2$) were incorporated at their injection concentrations. The rest
settings and inputs remain unchanged. To access $HO_2$ interference caused by the
ambient $RO_2$ conversion, the model underwent a three-days spins-up to simulate the
ambient $RO_2$ concentration. Then by adding the injected NO and $SO_2$ concentration to
the model, the conversion of $RO_2$ to OH in the inlet during $HO_2$ measurement mode
can be estimated.
**3. Results and Discussion**
**3.1 Results from Observations**
**3.1.1 Overview**
Figure 3 illustrates a time series showing observed concentrations of radical and
trace gases, along with meteorological parameters, from 12 November to 19 December
2022. In November, the conditions were characterised by warm temperatures ranging
from 29°C to 19°C and high relative humidity averaging 86%. In contrast, December
witnessed a significant decrease in temperature (ranging from 20°C to 9°C) and a
reduction in relative humidity (averaging 72%). Wind speeds during the campaign were
generally low, averaging 0.9 ± 0.6 m/s and typically remaining below 3.0 m/s, with
higher speeds occurring towards the end of December. In November, daytime winds
predominantly blew from the south, while nighttime winds came from the north. In
December, northerly winds predominated both day and night. Detailed hourly wind
speed and direction data are illustrated in Figure 3, and wind roses are shown in Figure
S3. On days with low wind speeds (< 0.5 m/s), $NO_X$ emissions from the G105 national
highway significantly influenced chemical measurements at the monitoring site,
causing morning NO levels to peak at several parts per billion (ppb). Isoprene
concentrations peaked in the afternoons, ranging from 0.2 to 1.7 ppb in November and
dropping to < 0.1 ppb in December. Other trace gases and particulate matter levels were
higher in November than in December.







**Figure 3** Time series of $HO_2$ and OH radicals between 12 November and 19 December, including measured weather conditions (temperature, RH, wind speed, and wind direction), primary sources of $HO_X$ radicals (ozone, HONO), important sinks of the radicals (CO, isoprene, and VOCs), and photolysis frequencies of $NO_2$ ($J_{NO2}$) and ozone ($J_{O1D}$). Non-continuous days during the campaign are delineated by a black line. The x-axis is in local time (+8 UTC).

Throughout the campaign, the daytime concentrations of OH and $HO_2$ consistently exceeded detection limits and showed distinct diurnal patterns. The OH concentrations typically peaked around midday, while the $HO_2$ levels reached their maximum approximately one to two hours later (Figure S4). The daily maximum concentration of OH varied significantly, ranging from $8.00 \times 10^6$ cm$^{-3}$ to nearly the detection limit of $2.54 \times 10^5$ cm$^{-3}$, with an average of $3.50 \pm 2.47 \times 10^6$ cm$^{-3}$ (Table 1). Similarly, the daily maximum concentration of $HO_2$ varied from $3.42 \times 10^8$ cm$^{-3}$ to $2.17 \times 10^7$ cm$^{-3}$, averaging $1.34 \pm 0.93 \times 10^8$ cm$^{-3}$ (Table 1). At nighttime, while the $HO_2$ levels generally remained above the detection threshold, the OH concentrations frequently approached the threshold. The average nighttime concentrations were $3.92 \times 10^7$ cm$^{-3}$ for $HO_2$ and $1.64 \times 10^5$ cm$^{-3}$ for OH. We compared the observed OH and $HO_2$ concentrations with those reported in previous studies conducted in urban, suburban, rural forest, and coastal sites. As illustrated in Figure S5, the OH concentrations were generally lower than those found in urban settings but similar to levels observed in suburban, rural, and forest environments. This suggests a moderate level of anthropogenic activity typical of mixed forest–rural settings. In contrast, the $HO_2$ concentrations during these periods were significantly lower than earlier observations in rural and forest environments, likely owing to reduced photochemical activity during these specific measurement times.

**Table 1** Average concentrations and standard deviation of measured species throughout the entire campaign (Total) and the selected 3 days cases from each cluster (PRD, CEC and CNC).



| Species (Unit) | Total | PRD | CEC | CNC |
|---|---|---|---|---|
| AveMax $OH_{Obs}$ $10^6$ $(cm^{-3})$ | 3.50±2.47 | 6.89±1.10 | 4.90±1.47 | 5.27±0.89 |
| $OH_{Obs}$ $10^6$ $(cm^{-3})$ | 0.93±1.49 | 1.60±2.18 | 1.36±1.61 | 1.19±1.77 |
| $OH_{DL}$ $10^6$ $(cm^{-3})$ | 0.52±0.34 | 0.43±0.28 | 0.44±0.22 | 0.90±0.55 |
| AveMax $HO_{2\ Obs}$ $10^8$ $(cm^{-3})$ | 1.34±0.93 | 2.32±1.25 | 2.36±0.92 | 1.82±1.02 |
| $HO_{2\ Obs}$ $10^8$ $(cm^{-3})$ | 0.59±0.51 | 0.76±0.63 | 1.10±0.68 | 0.67±0.55 |
| $HO_{2\ DL}$ $10^8$ $(cm^{-3})$ | 0.19±0.11 | 0.17±0.10 | 0.25±0.08 | 0.26±0.15 |
| Pressure (hpa) | 995±4 | 992±1 | 992±1 | 995±2 |
| Temp (°C) | 16±6 | 23±3 | 23±2 | 14±3 |
| RH (%) | 78±15 | 87±11 | 86±10 | 81±9 |
| Wind Speed (m/s) | 0.9±0.7 | 0.5±0.3 | 0.6±0.3 | 0.9±0.5 |
| $j_{O1D}$ $10^{-6}(s^{-1})$ | 3.2±5.4 | 3.5±6.0 | 3.6±5.9 | 4.0±6.6 |
| $j_{NO2}$ $10^{-3}$ $(s^{-1})$ | 1.3±1.9 | 1.3±2.1 | 1.4±2.0 | 1.6±2.3 |
| $j_{H2O2}$ $10^{-6}$ $(s^{-1})$ | 1.0±1.5 | 1.0±1.6 | 1.1±1.6 | 1.2±1.8 |
| $j_{NO3\ M}$ $10^{-3}$ $(s^{-1})$ | 4.0±6.4 | 4.2±6.8 | 4.3±6.7 | 5.3±7.9 |
| $j_{NO3\ R}$ $10^{-2}$ $(s^{-1})$ | 3.1±5.0 | 3.3±5.3 | 3.4±5.2 | 4.1±6.1 |
| $j_{HCHO\ M}$ $10^{-6}$ $(s^{-1})$ | 5.2±7.9 | 5.5±8.5 | 5.6±8.3 | 6.4±9.5 |
| $j_{HCHO\ R}$ $10^{-6}$ $(s^{-1})$ | 4.2±6.6 | 4.5±7.1 | 4.6±7.0 | 5.3±8.0 |
| $j_{HONO}$ $10^{-3}(s^{-1})$ | 0.2±0.3 | 0.2±0.4 | 0.2±0.3 | 0.3±0.4 |
| $j_{H2O2}$ $10^{-6}(s^{-1})$ | 1.0±1.5 | 1.0±1.6 | 1.1±1.6 | 1.2±1.8 |
| HONO (ppb) | 0.2±0.1 | 0.3±0.1 | 0.2±0.1 | 0.1±0.0 |
| $SO_2$ (ppb) | 0.6±0.8 | 0.5±0.7 | 0.4±0.5 | 0.4±0.5 |
| $NO_2$ (ppb) | 4.9±2.4 | 6.3±2.5 | 4.8±2.2 | 4.5±2.0 |
| NO (ppb) | 0.6±0.9 | 0.7±1.1 | 0.7±1.0 | 0.7±0.9 |
| CO (ppb) | 557±225 | 739±154 | 465±74 | 513±22 |
| Ozone (ppb) | 25±14 | 32±23 | 24±13 | 19±9 |
| Particle Count $(cm^{-3})$ | 1629±1303 | 2757±567 | 1723±836 | 1329±1304 |
| Particle Surface Area $(um^2/cm^3)$ | 86±72 | 186±51 | 84±28 | 48±19 |
| MACR (ppb) | 0.06±0.06 | 0.12±0.06 | 0.11±0.07 | 0.03±0.01 |
| *BVOCs (ppb) | 0.14±0.22 | 0.33±0.40 | 0.26±0.25 | 0.05±0.04 |
| *OVOCs (ppb) | 2.20±1.10 | 3.20±2.30 | 1.70±0.38 | 1.70±0.33 |
| *AVOCs (ppb) | 8.30±3.20 | 9.70±5.00 | 6.90±1.80 | 6.80±0.86 |

Notes: Concentrations are expressed in parts per billion (ppb) unless otherwise specified. Total
VOCs concentrations are categorized by origin (AVOCs, BVOCs, and OVOCs). For the average
concentration of each measured VOCs, refer to Table S4.
Figure 4 illustrates the results of the 24-hour backward trajectory analysis,
revealing three distinct but sequentially occurring phases. In the initial phase (Figure
4a), air masses originated from the urban and industrial zones of the Pearl River Delta
(PRD). This phase was characterised by intense photochemical activity, with ambient
temperatures exceeding 20°C and relative humidity levels surpassing 70%. During this
period, notably high concentrations of VOCs, ozone, and radicals were observed, with
the average daily maximum concentrations of OH and $HO_2$ radicals reaching $6.50 \pm$
$1.19 \times 10^6$ $cm^{-3}$ and $2.20 \pm 0.27 \times 10^8$ $cm^{-3}$, respectively. The subsequent phase was
characterised by air masses originating from Central East China (CEC, Figure 4b). This
phase showed reduced photochemical reactivity and lower concentrations of the



measured trace gases. The average daily maximum concentrations of OH and $HO_2$
during this phase were $4.35 \pm 2.19 \times 10^6$ cm$^{-3}$ and $1.96 \pm 0.90 \times 10^8$ cm$^{-3}$, respectively.
The final phase was influenced by air masses from Central North China (CNC, Figure
4c), which exhibited the lowest concentrations of trace gases and the least pronounced
average daily maximum concentrations in OH and $HO_2$ concentrations, measured at
$2.23 \pm 1.95 \times 10^6$ cm$^{-3}$ and $7.63 \pm 7.66 \times 10^7$ cm$^{-3}$, respectively. This phase coincided
with an increase in cloudy days and a decrease in temperatures, indicating reduced
photochemical conditions.

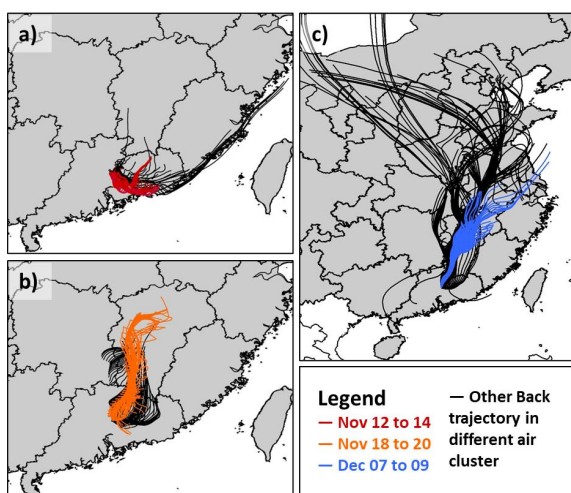

**Figure 4** 24-hour back trajectories for (a) Pearl River Delta (5 days), (b) Central East China (4 days),
and (c) Central North China (14 days) cases. Three days selected from each cluster for model
simulation are distinguished by different colours.

### 3.1.2   Selection of Cases

For each phase, a representative three-day period was selected for detailed analysis
based on the availability of comprehensive data (colored trajectories in Figure 4). In the
subsequent analysis, 'PRD,' 'CEC,' and 'CNC' refer to the selected periods
corresponding to the air masses originating from these regions. The average daily
maximum concentrations of OH and $HO_2$ radicals for these periods are presented in
Table 1. The average daily max OH concentrations were $6.89 \pm 1.10 \times 10^6$ cm$^{-3}$ in PRD,
$4.90 \pm 1.47 \times 10^6$ cm$^{-3}$ in CEC, and $5.27 \pm 0.89 \times 10^6$ cm$^{-3}$ in CNC, with a pronounce
decrease from PRD to CEC (of $1.99 \times 10^6$ cm$^{-3}$). The average daily max $HO_2$



concentrations were $2.32 \pm 1.25 \times 10^8$ cm$^{-3}$ in PRD, $2.36 \pm 0.92 \times 10^8$ cm$^{-3}$ in CEC,
and $1.82 \pm 1.02 \times 10^8$ cm$^{-3}$ in CNC, with a slight increase of $0.04 \times 10^8$ cm$^{-3}$ from PRD
to CEC and a more substantial drop of $0.54 \times 10^8$ cm$^{-3}$ from CEC to CNC. These trends
suggest a declining atmospheric oxidation capacity from PRD to CNC, with the factors
influencing OH radicals differing significantly between PRD and CEC, and those
affecting $HO_2$ radicals being more pronounced in CEC than CNC.
The precursor concentrations and meteorological parameters also varied across
cases in terms of statistics (Table 1) and diurnal variations (Figure S6). In the PRD case,
the average concentrations are characteristic of a rural environment, with AVOCs at
9.70±5.00 ppb, OVOCs at 3.20±2.30 ppb, BVOCs at 0.33±0.40 ppb, $NO_2$ at 6.3±2.5
ppb, and NO at 0.7±1.1 ppb. The NO concentration was affected by traffic sources as
no other fresh emission source nearby and the NO diurnal variation show a morning
peak in all three cases (Figure S6). In the CEC case, a general reduction in
anthropogenic influence is evident. AVOCs, OVOCs and $NO_2$ drop significantly to
6.90±1.80 ppb, 1.70±0.38 ppb, and 4.8±2.2 ppb respectively. Meanwhile, BVOCs and
NO remain close to PRD levels at 0.26±0.25 ppb and 0.7±1.0 ppb. In the CNC case,
the air mass is more aged with reduced biogenic emissions, reflected in further
decreases in BVOCs and $NO_2$ to 0.05±0.04 ppb and 4.5±2.0 ppb, respectively, due to
colder weather conditions. The temperature decreased significantly from PRD to CNC,
whereas the average peak photolysis frequency was comparable between PRD and
CNC, as shown in Table 1.
**3.2 Chemical budgets of OH and $HO_2$**
To investigate the OH and $HO_2$ chemical budgets during the three distinct periods,
we employed a box model constrained by observed concentrations of $NO_X$, VOCs, and
relevant meteorological parameters in the selected cases (base scenario which OH and
$HO_2$ concentrations were not constrained by observation here). The resulting OH and
$HO_2$ budgets, displaying typical bell-shaped patterns, are illustrated in Figure 5. During
midday (10:00–15:00), the main source of $HO_2$ was the recycling of RO species, with
rates of 3.22 ppb h$^{-1}$ for PRD, 2.09 ppb h$^{-1}$ for CEC, and 1.08 ppb h$^{-1}$ for CNC.





Additionally, HCHO photolysis contributed 0.75 ppb h$^{-1}$, 0.46 ppb h$^{-1}$, and 0.26 ppb
h$^{-1}$ for PRD, CEC, and CNC, respectively. The sinks of HO$_2$ varied among the cases,
mainly driven by radical termination mechanisms. The rate of radical self-reactions
decreased from PRD to CNC. In contrast, NO$_X$-radical reactions between CEC and
CNC were comparable, with respective rates of 0.39 ppb h$^{-1}$, and 0.33 ppb h$^{-1}$,
indicating a shift in radical termination mechanisms.

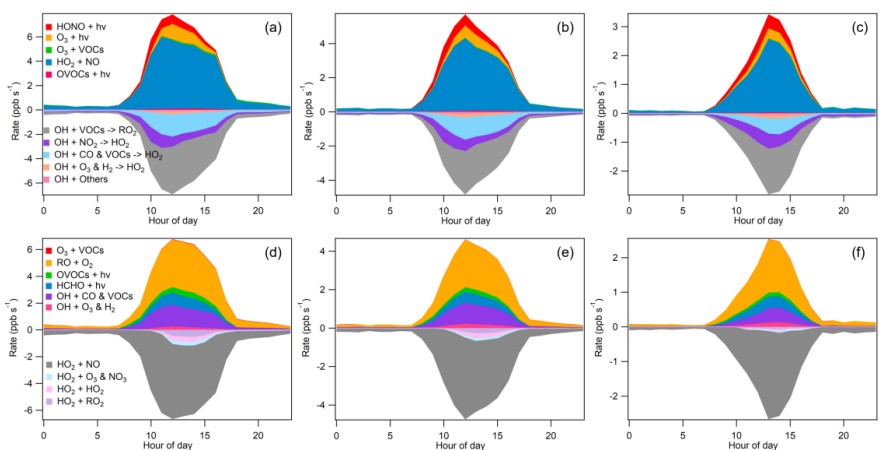

**Figure 5** Chemical budgets of OH and HO$_2$ for PRD (a, d), CEC (b, e), and CNC (c, f) simulated
using a chemical box model.
OH formation was predominantly driven by the HO$_2$ + NO reaction, contributing
5.18 ppb h$^{-1}$, 3.51 ppb h$^{-1}$, and 1.81 ppb h$^{-1}$ for PRD, CEC, and CNC, respectively.
Additionally, contributions from ozone photolysis and HONO increased from PRD to
CEC and then to CNC, with rates of 21.4%, 22.7%, and 24.6%, respectively. The
primary sinks for OH included reactions with VOCs to produce RO$_2$, with rates of 3.31
ppb h$^{-1}$, 2.02 ppb h$^{-1}$, and 1.13 ppb h$^{-1}$, and reactions with CO and other VOCs to
generate HO$_2$, contributing 1.55 ppb h$^{-1}$, 1.06 ppb h$^{-1}$, and 0.38 ppb h$^{-1}$ for PRD, CEC,
and CNC, respectively. These findings highlight the critical role of OH + VOC
reactions in the chemical budget of OH.
**3.3 Model vs. Observation**
To evaluate the performance of the MCM box model in simulating radical
chemistry, we compared the modeled and observed concentrations of OH and HO$_2$





radicals. In the CNC case, the model showed good agreement with observations for
both OH and HO$_2$ (Figures 6c and 6f). For the CEC case (Figures 6b and 6e), the model
moderately overestimated both radicals. In the PRD case which is the most polluted
and warmest among the three cases, the OH concentration was only slightly
overestimated, but the HO$_2$ concentration was substantially over-predicted by the
model (Figures 6a and 6d).
As mentioned in the introduction, the overestimation of OH and HO$_2$ radicals have
been reported other studies. Griffith et al., (2013) found that while modeled OH
concentrations agreed with measurements at a forested site, HO$_2$ concentrations were
overestimated. Similarly, (Kanaya et al., 2012; Bottorff et al., 2023) reported
simultaneous overestimations of OH and HO$_2$ in two rural sites. In our case, when the
model was constrained by observed OH concentrations, the overestimation of HO$_2$ was
resolved in the CEC case but remained largely unchanged in the PRD case (Figure S7).
The exact reasons for the model's overestimation of HO$_2$ (in PRD and CEC) and OH
(in CEC) in the remain unclear.

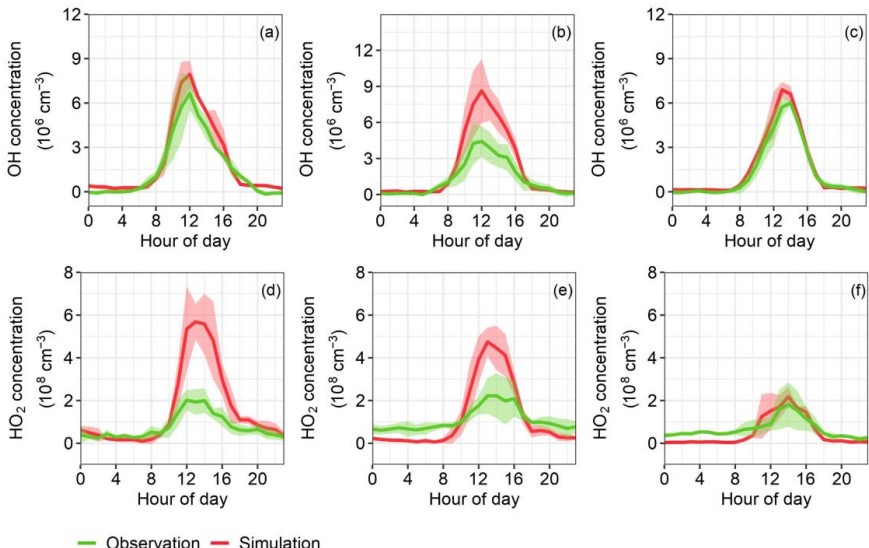

**Figure 6** Average diurnal variations of OH and HO$_2$ for PRD (a, d), CEC (b, e), and CNC (c, f)
from observational and modeling results. The solid line represents the average value, while the
shaded area indicates one standard deviation. The green line represents the observational results, the





red line shows the modeled results without constraining the observed $HO_2$ concentration (base
scenario).
**4.  Implication for model overestimation of $HO_X$**

4        OH and $HO_2$ are key oxidants that determine the atmosphere's oxidative capacity.

Inaccurate modelling of their sinks can lead to significant overestimation of this
capacity, resulting in skewed assessments of the impact of $HO_X$ on air pollution and
climate change. This problem is particularly pronounced in the case of ozone, a
widespread photochemical pollutant. To demonstrate this issue, we compared
simulation results from two modelling scenarios: the first scenario was constrained by
all observational parameters except OH and $HO_2$ (as described in section 3.2), while
the second scenario included constraints from all observational parameters, including
OH and $HO_2$ measurements.

13        As illustrated in Figure 7, not constraining free-radical measurement data in the

chemical model (the red line) led to overestimates of ozone's photochemical production
rates. In the PRD case, simulated midday $O_X$ ($O_3 + NO_2$) formation rates were
overestimated by 59.1% on average and 56.8% at peak values. In the CEC case, the
overestimation was 98.0% on average and 91.3% at peak $O_X$ rates, while the CNC case
exhibited the smallest overestimation, 52.4% on average and 25.8% at peak values.

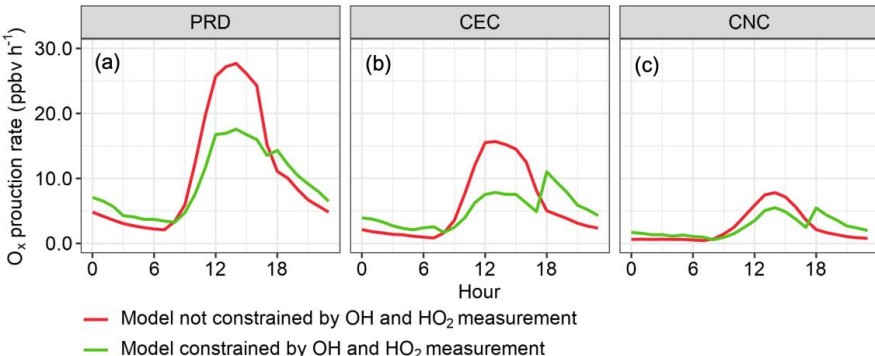

**Figure 7** $O_X$ ($O_3+NO_2$) photochemical production rates in three comparative cases: (a) PRD, (b)
CEC, and (c) CNC. The red lines represent rates modelled with constraints on all observed data
except OH and $HO_2$, while the green lines include constraints on all data, including OH and $HO_2$.





The overestimation of HO$_X$ also significantly affected the simulated concentration
of nitric acid (HNO$_3$), which is crucial for new particle formation and growth (Wang et
al., 2020). Figure 8 illustrates that the chemical model drastically overestimated nitric
acid production rates without constraints of free-radical measurements (the orange line).
The midday production rates of nitric acid were overestimated by factors of 3.16, 2.02,
and 3.41 in the PRD, CEC, and CNC cases, respectively. Such overestimations can
considerably impact assessments of air pollution and climate change.

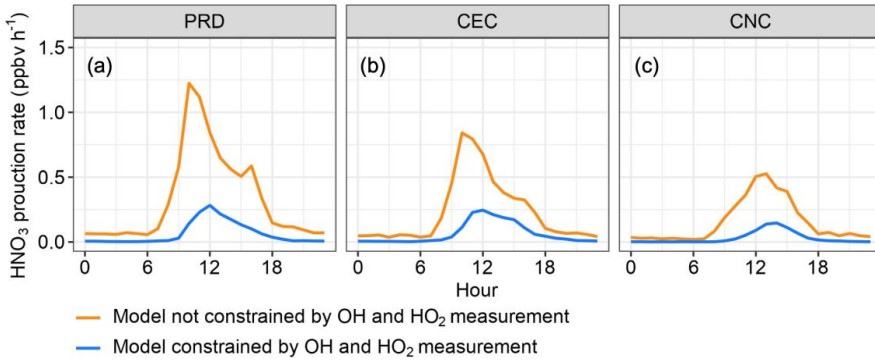

**Figure 8** Modelled HNO$_3$ concentrations in three comparative cases: (a) PRD, (b) CEC, and (c)
CNC. The red lines represent rates modelled with constraints on all observed data except OH and
HO$_2$, while the green lines include constraints on all data, including OH and HO$_2$.
**5. Conclusion**
HO$_2$ and OH concentrations were measured using a chemical ionization mass
spectrometer at a subtropical rural site in southern China from 12 November to 19
December 2022. The measurements indicated generally lower concentrations of OH
and HO$_2$ than those observed in previous studies at various sites. Backward trajectory
analysis revealed three distinct phases characterised by sequentially decreasing
pollution levels and temperatures. During the cold, clean period, model simulations
closely matched the observed OH and HO$_2$ concentrations. However, during the warm,
polluted period, the models overestimated both radicals. The over-prediction of HO$_X$
resulted in significant overestimations of the production rates of other secondary
pollutants such as ozone and nitric acid at the site. This study provides additional
evidence for current incomplete understanding of the HO$_X$ sources or sinks and calls





for more research to resolve the model–observation mismatch found in this work and
previous studies. It is critical to evaluate the capability of OH and $HO_2$ simulations in
major chemical transport models and Earth system models as inaccurate simulations of
OH and $HO_2$ may misguide the development of air pollution and global warming
control strategies.
**Data availability.** All of the data used to produce this paper can be obtained by
contacting Tao Wang (tao.wang@polyu.edu.hk).
**Supplement.** The online supplement for this article is available at:
**Author contributions.** TW conceived the $HO_x$ research. TW, XW and YZ planned and
organized the overall field campaign at Conghua. ZZ conducted the OH measurements
using CIMS, with contributions from TW and WS, QC, and SH. XF, ZR, XL, AG, QW,
CP, and XT performed the $JNO_2$ VOCs and OVOCs measurements. ZZ performed the
chemical box modelling with contributions from TC and QC. TC, ZZ, and TW analysed
the data and interpreted the result (ZZ analysed the time series and diurnal variations of
observation data; TC interpreted the results of box model, investigated the missing
sources; TW supervised and guided these processes). TC, ZZ, and TW wrote the paper.
All of the authors reviewed and commented on the paper.
**Competing interests.** One author (Tao Wang) is a member of the editorial board of
Atmospheric Chemistry and Physics. The authors have no other competing interests to
declare.
**Acknowledgments**
We thank David Tanner, Dr. Wei Pu, and Dr. Weihao Wang for developing the PolyU-
CIMS. We are also grateful to the Guangzhou Institute of Geochemistry, Chinese
Academy of Sciences, for providing access to its station and data on trace gases.
**Financial support.**
This research was financially supported by the Hong Kong Research Grants Council
(T24-504/17-N and 15223221 to Tao Wang), the National Science Foundation of China



(42293322 to Tao Wang), and The Hong Kong Polytechnic University Postdoc
Matching Fund Scheme (P0043403 to Tianshu Chen).

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
