# Peer review of "radicals at a subtropical rural site and implications for"

_EGUsphere, 2024_

## Author Comment (AC1)

**Response to Reviewers' Comments on Manuscript ID egusphere-2024-3210:
"Observation and modelling of atmospheric OH and HO₂ radicals at a
subtropical rural site and implications for secondary pollutants"**

We sincerely appreciate the editor and the three referees for thoughtful comments and valuable suggestions which has helped us to improve our manuscript. In response to your comments, we have undertaken a comprehensive revision of the manuscript.

Please find our itemized responses below and revisions in the re-submitted files. We use *italicized* text for your questions, **blue** for our response, and **red** to indicate where changes have been made in the manuscript. The changes inside the manuscript will be highlighted in yellow.

*Reviewer 1*

*The study presents valuable measurements of OH and HO₂ radicals using the CIMS technique at a subtropical rural site in southern China during November and December 2022. The data suggest generally lower concentrations of OH and HO₂ compared to previous studies. By categorizing the data into three distinct cases based on backward trajectory analysis, the study highlights good agreement between observations and model predictions under cold, clean conditions but significant overestimation under warm, polluted conditions. This overestimation, which affects secondary pollutant production, underscores the need for further investigation into $HO_x$ sources and sinks to resolve model-observation discrepancies. While this study provides valuable insights into $HO_x$ radical behavior in a subtropical rural environment, several aspects require clarification and deeper discussion. Addressing these issues will strengthen the study's conclusions and enhance its contribution to understanding radical chemistry and secondary pollutant formation in rural and polluted environments. Below are detailed comments.*

**Response:** Thanks very much for the constructive comments. Following your suggestions, we have revised the manuscript.

**Q1:** *The introduction section needs more comprehensive referencing. Important studies in the field of HOx radical chemistry, particularly those relevant to subtropical and rural environments, should be cited to provide better context and demonstrate the study's relevance.*

**Response:** We have incorporated additional references on $HO_x$ radical chemistry in subtropical and rural environments into the introduction, results and discussion sections. Additionally, we have revised some descriptions to better provide context and highlight the relevance of our study.

Revision on pages 2-3: "Previous studies have shown that models generally predict OH levels well in polluted conditions (NO > 1 ppb), but notable overestimation were observed under low NO and aged conditions, such as coastal areas (Kanaya et al., 2007; Zou et al., 2023), marine boundary layers (Berresheim et al., 2002; Carslaw et al., 1999), and the rural area (Bottorff et al., 2023; Kanaya et al., 2012). Missing OH sinks from

both measurement or chemical mechanism were proposed as the primary reason for the overestimation (Lou et al., 2010; Yang et al., 2016; Hansen et al., 2014 Thames et al., 2020). Underestimation of OH concentrations were also observed in high biogenic VOCs (BVOCs) and low NO (<1 ppb) conditions which generally happen in the subtropical or tropical area (Hofzumahaus et al., 2009; Lelieveld et al., 2008; Tan et al., 2001; Whalley et al., 2011)……Those results called for more measurement and modelling in the subtropical and tropical rural areas. ”

Revision on page 16: “The modeled and observed concentrations of OH and $HO_2$ radicals were compared to evaluate the performance of the model. In the PRD case (Figure 7), which is the most polluted and warmest among the three cases, the OH concentration was only slightly overestimated, whereas the $HO_2$ concentration was substantially overpredicted by the model during the daytime. Similar result has been observed at another rural site (Kanaya et al., 2012). For the CEC case (Figure 8), the model moderately overestimated both radicals during the daytime but underestimated the nighttime $HO_2$ concentration, which is similar to the findings at a rural forest site (Bottorff et al., 2023). In the CNC case (Figure 9), the model results were generally within the measurement uncertainty, with some daytime overestimation of $HO_x$ on December 7 (similar to the PRD case) and nighttime underestimation of $HO_2$ (similar to the CEC case). In the following section, we conduct sensitivity tests to explore the possible reasons for the model observation discrepancy in the PRD and CEC cases.”

Revision on page 18 line 9: “This result suggests that aligning the modeled OH and $HO_2$ with observations may require introducing a strong, unknown process for $HO_2$ that efficiently recycles OH with a high yield (Kanaya et al., 2012).”

**Q2:** *The calibration procedures for OH and $HO_2$ require further clarification. Given the critical role of the calibration factor in determining OH concentrations, a detailed explanation of the calibration methodology is essential.*

**Response:** We have added more details on OH and $HO_2$ calibration to the Supplementary Information (SI).

Revision in Text S1 on page 8 of the SI: “The calibration of Chemical Ionization Mass Spectrometer (CIMS) involves the generation of OH and $HO_2$ radicals through photolysis of water vapor by 184.9 nm light, as outlined in Reaction R30. The concentration of radicals produced during calibration is determined from the known concentration of water vapor $[H_2O]$, which is calculated from water vapor pressure and the relative humidity and temperature. Other essential parameters include the photolysis cross-section of water vapor ($\sigma_{H2O}$ = 7.14 × 10$^{-20}$ cm$^2$; Cantrell et al., 1997), the photolysis quantum yield ($\Phi$, assumed to be 1, Kürten et al., 2012) and the photon flux (*It* value, see details about *It* value determination on Kürten et al., 2012). The generated radical concentrations ($[OH]$ and $[HO_2]$) are calculated using the following equations:

$$[OH] = [HO_2] = [H_2O] * \sigma_{H_2O} * \Phi * It$$

From these values, the calibration factors for OH and $HO_2$ ($C_{OH}$ and $C_{HO2}$) are calculated using the signals obtained during calibration ($S_{OHcal}$ and $S_{HO2cal}$), as expressed in the transformed equations E1 and E2:

$$C_{OH} = \frac{1}{[OH]_{cal}} \times \frac{S_{OHcal}}{S_{62}} \text{ (E1, transformed)}$$

$$C_{HO2} = \frac{1}{[HO_2]_{cal}} \times \frac{S_{HO2cal}}{S_{62}} \text{ (E2, transformed)}$$

The calibrator produced OH and $HO_2$ concentrations in the range of $3 \times 10^7$ to $1 \times 10^9$ $cm^{-3}$ depending on RH conditions in 10 LMP synthetic air. The more detailed information on calculation procedures is given in our previous study (Zou et al., 2023).

**Q3:** *The manuscript should specify the scavenge efficiency of OH during the measurement process. How was it ensured that OH radicals were entirely removed? This information is crucial to validate the reported OH concentrations.*

**Response:** The scavenge efficiency of OH is 100% in our system. This is achieved by increasing the scavenger gas ($C_3F_6$) amount to the level at the OH signal stopped decreasing during calibration. Any remaining signals are attributable to instrument noise, which includes potential contributions from Criegee intermediates and ambient sulfuric acid, rather than residual unscavenged radicals.

Besides, the terminology of "scavenge efficiency" we used in Table S4 may unintentionally implies incomplete removal of OH and $HO_2$ radicals. We have revised them to "background to signal ratio (B/S Ratio)" to more accurately reflect that the residual signals are measures of background noise rather than incomplete scavenging.

Revision in Text S3 on page 9 line 15 of the SI: "To determine the amount of $C_3F_6$ that is needed to achieve complete OH scavenge, we gradually increased $C_3F_6$ added to high concentrations of OH and $HO_2$ ($[HO_x] \approx 10^9$ $cm^{-3}$) generated from the calibrator in synthetic air until no further reduction in the measured signal, which indicates complete scavenging of OH. This point defines the background noise which is attributed to any Criegee intermediates and ambient sulfuric acid."

**Q4:** *The efficiency of $HO_2$ conversion via its reaction with NO should be discussed in detail. Is the conversion complete, or is it assumed to operate at a constant efficiency? This factor significantly affects the accuracy of $HO_2$ measurements.*

**Response:** Based on the laboratory tests, the conversion of $HO_2$ to OH by NO and further to $H_2SO_4$ for detection is 100% ($[H_2SO_4]/[HO_2] = 1$). We assume similar complete (and constant) conversion in the ambient air because the added conversion gases overwhelm the ambient concentrations (~500 times higher) in our study site.

Revision in Text S2 on page 10 of the SI: "

"During $HO_2$ measurement, ambient $HO_2$ converted from NO to OH. It should be noted that in $HO_2$ mode, the increasing NO concentration can enhance $HO_2$ conversion to OH

(R11), but excessive NO levels trigger the HONO formation when reacts with OH (R15), competing with the OH conversion process by $SO_2$ (R21) and lowering the detection efficiency for OH. Consequently, the NO to $SO_2$ concentration ratio is crucial for $HO_2$ measurements. Sensitivity tests revealed an optimal $[NO]/[SO_2]$ ratio of 0.1 for the PolyU-CIMS and 100% conversion of $HO_2$ in the laboratory ($[H_2SO_4]/[HO_2] = 1$), aligning with prior research recommendations (Edwards et al., 2003; Sjostedt et al., 2007). Because the concentrations of both $SO_2$ and NO injected to sample flow are maintained at levels over 100-1000 times higher than those in the ambient atmosphere and the injection flow rates are fixed, the efficiency of the $HO_2$ to OH conversion remains stable and is believed to be at completion."

**Q5:** *The manuscript should elaborate on the $HO_2$ titration process, specifically how $HO_2$ is converted to OH and subsequently reacts with $C_3F_6$. The scavenge efficiency for this step may differ from that of OH, which could influence the accuracy of $HO_2$ measurements.*

**Response:** As responded to Q4, $HO_2$ was converted to OH at 100% by adding high concentrations of NO. The $HO_2$ converted OH was scavenged by $C_3F_6$ in the same way as for the OH mode (see the response to Q3). The $C_3F_6$ concentration that achieved 100% scavenging of OH from $HO_2$ conversion was adopted in both OH and $HO_2$ modes to ensure complete scavenging OH in the two modes.

To further elaborate on the background measurement process, we have added the following detailed description in Text S3 on page 10 of SI the revised manuscript:

"In the background mode, scavenger gases $C_3F_6$ are introduced into the sample flow along with $SO_2$. Given that the concentration of $C_3F_6$ is 100 times higher than that of $SO_2$, the ambient OH and any ambient $HO_2$ converted to OH are scavenged by $C_3F_6$, rather than being converted to $H_2SO_4$ for further detection. As a result, in background mode, we are able to accurately determine the interference signals."

**Q6:** *The unit of the calibration factor should not be expressed as 'cm$^{-3}$.' This error needs correction for consistency and clarity.*

**Response:** The appropriate unit should be cm$^3$ this has now been corrected throughout the manuscript.

**Q7:** *The manuscript claims a measurement accuracy of 44% for $HO_2$ and 46% for OH. However, this discrepancy is counterintuitive and should be explained, as $HO_2$ measurements are typically less accurate than OH measurements.*

**Response:** The reported accuracies for OH and $HO_2$ were mistakenly reversed in the initial submission. This has been corrected in the revised manuscript, with 44% for OH and 46% for $HO_2$.

**Q8:** *Previous studies in rural areas generally report underestimation of OH and $HO_2$, yet this study finds significant overestimation under certain conditions. This*

*discrepancy requires further discussion, especially concerning the chemical mechanisms and environmental factors leading to such outcomes.*

**Response:** The previous studies found instances of good agreement, overestimations and underestimations in similar studies. We have summarized these discrepancies in the Introduction and Discussion section of the revised manuscript and gives detailed discussion on possible explanations on section 3.3.1 to 3.3.3 respect to different scenarios. For the revisions on the Introduction section please refer to Q1.

Rewrite discussion on page 16:

"The modeled ……Similar result has been observed at another rural site (Kanaya et al., 2012). For the CEC case (Figure 8), the model moderately overestimated both radicals during the daytime but underestimated the nighttime $HO_2$ concentration, which is similar to the findings at a rural forest site (Bottorff et al., 2023). In the CNC case (Figure 9), the model results were generally within the measurement uncertainty, with some daytime overestimation of $HO_x$ on December 7 (similar to the PRD case) and nighttime underestimation of $HO_2$ (similar to the CEC case). In the following section, we conduct sensitivity tests to explore the possible reasons for the model observation discrepancy in the PRD and CEC cases."

Additional figures and discussion on page 17-19:

[Figure]

"**Figure 7** observed and simulated time series of OH and $HO_2$ for the PRD case. The "Obs" subscript denotes the observation data. "Base" denotes the result of Baseline scenario as described on Box Model section. "Cons" denotes the results with additional constrained species compare to "Base". "2.5 × $K_{HO2+NO}$" denotes the results with increasing the reaction rate coefficient of R11 by a factor of 2.5

[Figure]

**Figure 8** observed and simulated time series of OH and HO$_2$ for the CEC case. The "Obs" subscript denotes the observation data. "Base" denotes the result of Baseline scenario as described on Box Model section. "Cons" denotes the results with additional constrained species compare to "Base". "Traffic" denotes the sensitivity test results with consideration of vehicular emission (see Test S6 in SI). "10 × [PANs] (21:00-05:00)" denotes the results with increasing nighttime PANs concentration by a factor of 10.

[Figure]

**Figure 9** observed and simulated time series of OH and HO$_2$ for the CNC case. "Base" denotes the result of Baseline scenario as described on Box Model section.

**3.3.1 Substantial overestimation of HO$_2$ in PRD case**

To explain the HO$_2$ over-simulation by the base model, we constrain OH or HO$_2$ and compared to the base case (without constraining OH and HO$_2$). Result shows that

constraining $HO_2$ causes the model to underestimate OH (blue line in Figure 7a), while constraining OH leads the model to still substantially overestimate $HO_2$ (blue line in Figure 7b). This result suggests that aligning the modeled OH and $HO_2$ with observations may require introducing a strong, unknown process for $HO_2$ that efficiently recycles OH with a high yield (Kanaya et al., 2012). A sensitivity analysis shows that increasing the reaction rate coefficient of $HO_2 + NO \rightarrow OH + NO_2$ (R11) by a factor of 2.5 would largely reduce both the $HO_2$ overestimation and the OH underestimation as shown by the black line in Figure 7. However, it is not clear what such OH cycling reaction is. Thus, the exact cause of the overestimation of $HO_2$ in the PRD case remain unresolved.

**3.3.2 Moderate overestimation of both OH and $HO_2$ radicals in CEC case**

Unlike the PRD case, constraining either OH or $HO_2$ in the CEC case generally reduces the daytime overestimation of both $HO_2$ and OH. These results indicate an additional sink for both OH and $HO_2$, as suggested by Bottorff et al. (2023). However, the OH concentration shows an overestimation in the morning when $HO_2$ was constrained, which may suggest missing OH reactivity in the morning. To further investigate the underlying causes, we examined the correlations between various pollutants. The significant negative correlation between CO and NO ($R^2$=0.49, p=0.01, Figure S6b) suggests that CEC in the morning may have been influenced by emission from fresh complete combustion during the CEC case, whereas such correlations for PRD and CNC are not significant (Figure S6a and c). This indicates that the missing OH reactivity of CEC in the morning is possibly related to fresh vehicle emissions.

Diesel vehicle exhausts are rich in OVOCs relative to total VOCs (Yang et al., 2023). In our study, OVOCs were measured, except formaldehyde and acetaldehyde. We conducted a sensitivity analysis by adding these two OVOCs in the model (see Text S7 for details). After accounting for their influence, the overestimation of OH in the morning with constraining $HO_2$ could be significantly reduced (Figure 8a black line).

**3.3.3 Nighttime underestimation of $HO_2$ in CEC case**

Ozone and $NO_3$ reactions with alkenes can produce $HO_2$ at night (Walker et al., 2015). In our study, alkenes are unlikely to be the main cause for the underestimation because the major alkenes were measured, and the alkenes concentrations in the CEC case were much lower compared to the PRD case in which no underestimation of nighttime $HO_2$ was found. A previous study (Whalley et al., 2010) showed that nighttime $HO_2$ underestimation at a clean tropical Atlantic site was significantly reduced by constraining the model with higher PAN. In our study, PAN was not measured. The model simulated nighttime PAN mixing ratios (0.1-0.7 ppb) were lower than previous observed nighttime results in the coastal (up to 1 ppb) (Xu et al., 2015) and mountain site (up to 2 ppb) (Wang et al., 2023) in southern China. To assess the impact of PAN

concentration on nighttime $HO_2$ levels, a sensitivity analysis was conducted in which the PAN concentration was increased. The results show that only when the PAN concentrations were increased by tenfold, the model simulated nighttime PAN level could match the observations (Figure 8b, black line). This suggests that underestimated PAN might have contributed to the model's nighttime $HO_2$ underestimation, but other processes must have a larger contribution."

Additional Figure in SI:

[Figure]

**Figure S6** Relationship between NO and CO concentrations in (a) PRD, (b) CEC and (c) CNC from 7:00 to 10:00. The solid lines depict the linear regression fit, with the corresponding equations $R^2$ and P values annotated on the plot.

Revised Text S7 on SI:

"A sensitivity test was conducted for the CEC case to account for the missing OH reactivity in the morning. This missing OH reactivity was attributed to unmeasured species in the fresh diesel exhaust. To estimate this, we first calculated the total OH reactivity of the exhaust based on the reactivity of $NO_x$ and CO, along with the diesel exhaust source profile. The contributions from $NO_x$ and CO were then subtracted. The remaining OH reactivity was allocated to formaldehyde and acetaldehyde, with their concentrations adjusted accordingly. This allocation was justified by the significant contribution of OVOCs to the total reactivity of diesel exhaust (Yao et al., 2015; Mo et al., 2016), as formaldehyde and acetaldehyde were not measured in this study. The sensitivity test was performed following these steps:

1. First, we calculated the OH reactivity of freshly emitted NOx and CO at each time step. We assumed that the pollutant concentrations at the time of the highest NO concentration did not undergo significant photochemical loss. For each time step, we calculated the ratio of the OH concentration at the time of the highest NO concentration to the OH concentration at that time step. This ratio was then multiplied by the OH reactivity of ambient $NO_x$ and CO at that time step to estimate the OH reactivity from the emitted $NO_x$ and CO.

2. The observed exhaust OH reactivity was determined by dividing the OH reactivity of emitted $NO_x$ and CO by 20%, which represents the minimum contribution of $NO_x$ and CO to the observed OH reactivity in exhaust in China (Yang, 2023).

3. The total exhaust OH reactivity was derived by dividing the observed exhaust OH reactivity by 60%, to account for the approximately 40% of OH reactivity missing in Chinese diesel exhaust (Yang, 2023). The OH reactivity of emitted $NO_x$ and CO was then subtracted from the total exhaust OH reactivity.

4. The remaining OH reactivity was allocated to formaldehyde and acetaldehyde in a 1:1.6 ratio, and their concentrations were adjusted accordingly. This ratio was calculated based on the concentration ratios of formaldehyde and acetaldehyde in diesel exhaust (Yao et al., 2015) and their respective OH reactivity coefficients.

**Q9:** *The number of significant figures reported for parameters should align with the detection limits of the instrument. Retaining two decimal places for all parameters, irrespective of their precision, is inconsistent.*

**Response:** We have adjusted all values aligning with the instrument's detection limits.

**Q10:** *The units in Figure 5 should be corrected from 'ppb/s' to 'ppb/h' for consistency and to align with the typical units used in radical budget analysis.*

**Response:** We have updated the units in Figure 6 (Figure 5 in previous version) from 'ppb/s' to 'ppb h$^{-1}$'.

**Q11:** *In the legend of Figure 5a, the reaction "OH + $NO_2$" should be correctly identified as forming $HNO_3$ rather than $HO_2$.*

**Response:** Thank you for pointing this out. We have revised the legend in Figure 5.

**Q12:** *The manuscript inconsistently classifies the measurement site. Although it is described as a rural site, Figure S5 attributes it to a forest environment. Furthermore, the observed BVOC concentrations are much lower than AVOC concentrations, indicating a rural rather than a forested environment. This ambiguity should be resolved for clarity.*

**Response:** We acknowledge the inconsistency in describing the measurement site. The revised manuscript will attribute the result to a rural site in Figure S5, and throughout the manuscript.

**Q13:** *The manuscript lacks an assessment of the model's performance in simulating key species such as ozone and OVOCs. Given the uncommon degree of $HO_x$ overestimation in a similar environment, evaluating the model's performance against observed concentrations of these species is necessary to validate the findings.*

**Response:** Our box model is constrained with both observed VOCs and OVOCs to calculate OH, and $HO_2$. The box model is not suitable for predicting Ozone and OVOC concentrations due to the lack of regional transport process. Because of the limited number of OVOCs measured in our study, it is likely our model underestimate OVOCs contribution to $HO_2$, which would lead to larger over-prediction of $HO_2$.

---

## Author Comment (AC2)

**Response to Reviewers' Comments on Manuscript ID egusphere-2024-3210:
"Observation and modelling of atmospheric OH and HO₂ radicals at a
subtropical rural site and implications for secondary pollutants"**

We sincerely appreciate the editor and the three referees for thoughtful comments and valuable suggestions which has helped us to improve our manuscript. In response to your comments, we have undertaken a comprehensive revision of the manuscript.

Please find our itemized responses below and revisions in the re-submitted files. We use *italicized* text for your questions, **blue** for our response, and **red** to indicate where changes have been made in the manuscript. The changes inside the manuscript will be highlighted in yellow.

**Reviewer 2**

*This manuscript details measurements of OH and HO₂ radicals made using a CIMS instrument at a subtropical rural site near the Pearl River Delta in China. These measurements are presented along with radical concentrations from a box model featuring the Master Chemical Mechanism v3.3.1. In general, when constrained to a suite of measured trace gases and meteorological parameters, the model overpredicted the measured radical concentrations, especially during warmer, more polluted periods. The authors also use an additional model, further constrained to the measured radical concentrations, to illustrate that the overprediction of HOₓ species results in a significant overestimate of the production rates of secondary pollutants such as ozone and nitric acid.*

*Accurate field measurements of OH and HO₂ are extremely challenging, and the results presented in this manuscript are important to characterizing radical chemistry and the overall oxidative capacity of the atmosphere. While it is clear that a significant amount of work went into the field measurements described in this study, a lack of detail regarding the instrumentation and calibration procedures and limited discussion surrounding the model overestimation of the measured HOₓ concentrations limits the conclusions that can be drawn from the results. The strength of the manuscript would be greatly improved if the authors expanded sections 3.2 and 3.3 and offered insight into potential explanations for the discrepancies between measured and modeled HOₓ concentrations.*

**Response:** Thank you for your thoughtful review, we have improved the sections on instrumentation, calibration and result discussion. We have responded to your comments in detail under each question.

*More specific comments are included below:*

**Q1:** *Section 2.2 – There are few details on the calibration procedure for OH and HO₂. Given the importance of the calibration factor for each species, a brief description of the process should be included. The authors do describe that equivalent amounts of OH and HO₂ are produced by the calibration source – is the residence time of radicals in the calibration source sufficient such that wall interactions and radical-radical loss*

*mechanisms must be considered to determine the concentrations of OH and HO₂ that exit the calibrator and enter the sampling inlet?*

**Response:**

Thanks for the comments. The detailed procedures on calibration were added to the SI and shown in the previous response (refer to referee 1 Q2). The calibration unit and setup are similar to the one developed by Kürten et al. (2012). In this calibration unit, the proximity of the calibration lamp to the sample inlet (less than 1 cm) and the swift transition of radicals from the front injector (1 cm away) (refer to Figure S6a as shown below) gives a residence time of less than 20 ms. This short residence time, combined with the laminar flow conditions in the calibrator and inlets (Reynolds Number < 2000), effectively negates the potential for significant radical-radical loss and wall interactions. Thus, the wall loss and radical-radical loss between source and conversion zone is believed negligible.

Revision in Text S4.1 on page 10-11 of the SI:

"Furthermore, the potential for radical-radical loss after radicals exit the calibrator and enter the sampling inlet was considered. Given the flow speed of the ambient inlet (12.2 m/s), the sample inlet (55 cm/s), and the distances involved—the calibration lamp is less than 1 cm from the sample inlet, and the sampling port to the front injectors is 1 cm—it can be calculated that the transport time for radicals to the front injectors for reactions is less than 20 ms. This brief transport time is sufficiently short to prevent significant radical-radical losses. Additionally, since the sample inlet draws the central part of the airflow within the ambient inlet, and the flow in the sample inlet is laminar, wall losses at this stage are also considered negligible."

Besides, triggered by the reviewer's comment on wall loss in the calibration unit, we also discussed possibility of loss of OH and HO₂ in the ambient inlet on Text S4.1 on page 10 line 6 of the SI, as below.

"Wall losses in the ambient inlet were evaluated by varying the distances of the calibration lamp from the inlet to assess potential effects on signal attenuation. The instrument was calibrated in two distinct configurations: initially, the lamp was positioned close to the CIMS sample inlet (Figure S6a, and subsequently, moved away from the CIMS sample inlet (Figure S6b. By comparing the observed signals from these two configurations, we were able to calculate the wall losses associated with the ambient inlet. The results indicated no significant difference (<1%) between the two measurements, suggesting negligible wall losses in the sampling system. "

[Figure]

Figure S8 the calibration process during ambient sampling in (a) close and (b) far positions.

**Q2:** *Section 2.2 and Table S3 – Similar to above, there are limited details regarding the timing of the measurement sequence and the addition of the scavenger gas. The manuscript should not describe the OH, HO₂, and H₂SO₄ measurements as simultaneous but should instead detail the amount of time spent in each measurement mode. In the main text, the authors also describe that OH and HO₂ concentrations are derived from a simple subtraction of background signals, while Table S3 lists the scavenging efficiency for OH and HO₂. How are these scavenging efficiency values determined and how are they factored into the determination of radical concentrations? Are there any lingering effects of the scavenger gas that must be considered similar to the residual NO that is described in Text S2? These details would provide additional confidence in the radical measurements.*

**Response:** Thank you for your inquiry. The duty cycle for our CIMS were 6 minutes for HO₂, 4 minutes for H₂SO₄, and 50 minutes for OH, alternating between 1 min measurement model and 1 min background mode. The scavenger efficiency for OH and HO₂ are 100%. The lingering effect of scavenger gas is negligible (see detail discussion below).

We have included a detailed duty cycle on page 13 to 14 in the Supplementary Information (Text S5, Tabel S6 and Figure S6) of the revised manuscript as below:

"As detailed in Section 2.2, the PolyU-CIMS was configured to sequentially measure HO₂, H₂SO₄, and OH within each hour during the field study, corresponding to changes in injection gases. Table S3 outlines the hourly schedule and injection gases Figure S9 a 1-hour duty cycle.

Table S6 Duty cycle and injection gases for targeted chemical analysis.

| Purpose | Measurement Mode | Signal 97 Label | Chemicals injected to the sample flow through different injectors | | Duty time (s) | Repeat times | Total Duration (mins) |
|---------|------------------|-----------------|-------------------------|------------------|---------------|--------------|------------------------|
| | | | Front Injectors | Rear Injectors | | | |
| $HO_2$ | SIG | $S_{97NO}$ | NO, $N_{2(p)}$, $SO_2$, | Sca, $Sca_{(p)}$ | 60 | 3 | 6 |
| | BKG | $S_{97NOSca}$ | NO, $Sca_{(p)}$, $SO_2$ | Sca, $N_{2(p)}$ | 60 | | |
| $H_2SO_4$ | SIG | $S_{97w/o}$ | - | Sca, $Sca_{(p)}$ | 60 | 2 | 4 |
| | BKG | $S_{97w/o}$ | - | Sca, $N_{2(p)}$ | 60 | | |
| OH | SIG | $S_{97SO2}$ | $N_{2(p)}$, $SO_2$, | Sca, $Sca_{(p)}$ | 60 | 25 | 50 |
| | BKG | $S_{97SO2Sca}$ | $Sca_{(p)}$, $SO_2$ | Sca, $N_{2(p)}$ | 60 | | |

Notes:

Front and Rear Injectors - The injector pairs as demonstrated in the Figure S2

SIG & BKG – the signal and background modes.

Sca - scavenger gases, $C_3F_6$ in this study.

$Sca_{(p)}$ - scavenger gases, add through the pulsed flow

$N_{2(p)}$- nitrogen gases, add through the pulsed flow

[Figure]

Figure S9 Variation of signal intensity at m/z 97 during a 1-hour duty cycle of CIMS measurement."

Regarding to referee's comment on lingering effects of scavenger gas, our tests showed it is negligible in our CIMS. Details added on page 9 line 20 of the SI in revised manuscript (Test S3 and Figure S7):

"In our setup, there is residual $C_3F_6$ present after CIMS switches from background to signal mode, but it does not affect the measurement results. As shown in Figure S7, after switching off $C_3F_6$, the measurement signals rapidly return to their initial levels within 20 seconds. Data affected by $C_3F_6$ residual are excluded to minimize the impact of the residual $C_3F_6$ on the measurements.

[Figure]

Figure S7 Variation of signal intensity m/z at 97 before $C_3F_6$ addition with time, after addition and switching off of $C_3F_6$ in synthetic air containing OH of $\sim 5 \times 10^8$ cm$^{-3}$."

**Q3:** *Page 8, Line 11. After this description of HO$_2$ uptake on aerosols, this process is not included as a loss mechanism in Figure 5 or discussed in the remainder of the manuscript despite SMPS measurements of particle size and number being shown in Figure 3. Is this uptake negligible compared to other loss mechanisms shown in Figure 5? Is RO$_2$ loss on aerosols also included in the model?*

**Response:** Thank you for your comment. The heterogeneous uptake of HO$_2$ on aerosols was included in our model with minor contribution to the loss of HO$_2$. But it was not shown in Figure 5 in the previous version. In the updated manuscript, we have revised this figure to include this process as a loss mechanism, indicating insignificant uptake of HO$_2$. We have clarified this in the updated manuscript. The losses of RO$_2$ on aerosols are not considered in our model.

Revised Figure 6 (Figure 5 in previous version) on page 15:

"

[Figure]

**Figure 6** Chemical budgets of OH and $HO_2$ for PRD (a, d), CEC (b, e), and CNC (c, f) simulated using a chemical box model."

Revised section 3.2 on page 15:

"The sinks of $HO_2$ varied among the cases with minor contribution from the uptake process, driven by radical termination mechanisms. The rate of radical self-reactions decreased from PRD to CNC."

**Q4:** *Page 16, Lined 7-15: As mentioned above, the significant overprediction of OH and $HO_2$during the PRD and CEC should be the main focus but the current manuscript offers very little in the form of discussion. I suggest expanding this section to include the rate of $HO_2$loss necessary to account for the difference between modeled and measured concentrations, how this rate compares to other processes shown in Figure 5, and potential explanations for the discrepancies.*

**Response:** We agree that the overestimation of radicals should be a key focus of the manuscript. Following the reviewer's suggestion, we reviewed the literature and conducted model sensitivity tests to explore potential causes. In brief, the overestimation of $HO_2$ at PRD may result from missing $HO_2$-to-OH conversion processes. For the CEC air mass, we hypothesize that the overestimation of OH and $HO_2$ is primarily due to radical terminal reactions and limited measurements of OVOCs in this air mass which was strongly influenced by vehicular emissions. The detail and lengthy discussions have been added in the revised version and made on pages 19–21 as shown below:

Revised section 3.3.1 to 3.3.2 on page 17 to 19:

"

[Figure]

**Figure 7** observed and simulated time series of OH and HO₂ for the PRD case. The "Obs" subscript denotes the observation data. "Base" denotes the result of Baseline scenario as described in Box Model section. "Cons" denotes the results with additional constrained species compared to Base. "2.5 × $K_{HO2+NO}$" denotes the results with increasing the reaction rate coefficient of R11 by a factor of 2.5.

[Figure]

**Figure 8** observed and simulated time series of OH and HO₂ for the CEC case. The "Obs" subscript denotes the observation data. "Base" denotes the result of Baseline scenario as described in Box Model section. "Cons" denotes the results with additional constrained species compared to Base. "Traffic" denotes the sensitivity test results with consideration of vehicular emission (see Test S7 in SI). "10 × [PANs] (21:00-05:00)" denotes the results with increasing nighttime secondary concentration of PAN by a factor of 10.

[Figure]

**Figure 9** observed and simulated time series of OH and HO$_2$ for the CNC case. "Base" denotes the result of Baseline scenario as described in Box Model section.

**3.3.1 Substantial overestimation of HO$_2$ in PRD case**

To explain the HO$_2$ over-simulation by the base model, we constrain OH or HO$_2$ and compared it to the base case (without constraining OH and HO$_2$). Result shows that constraining HO$_2$ causes the model to underestimate OH (blue line in Figure 7a), while constraining OH leads the model to still substantially overestimate HO$_2$ (blue line in Figure 7b). This result suggests that aligning the modeled OH and HO$_2$ with observations may require introducing a strong, unknown process for HO$_2$ that efficiently recycles OH with a high yield (Kanaya et al., 2012). A sensitivity analysis shows that increasing the reaction rate coefficient of HO$_2$ + NO → OH + NO$_2$ (R11) by a factor of 2.5 would largely reduce both the HO$_2$ overestimation and the OH underestimation as shown by the black line in Figure 7. However, it is not clear what such OH cycling reaction is. Thus, the exact cause of the overestimation of HO$_2$ in the PRD case remains unresolved.

**3.3.2 Moderate overestimation of both OH and HO$_2$ radicals in CEC case**

Unlike the PRD case, constraining either OH or HO$_2$ in the CEC case generally reduces the daytime overestimation of both HO$_2$ and OH. These results indicate an additional sink for both OH and HO$_2$, as suggested by Bottorff et al. (2023). However, the OH concentration shows an overestimation in the morning when HO$_2$ was constrained, which may suggest missing OH reactivity in the morning. To further investigate the underlying causes, we examined the correlations between various pollutants. The significant negative correlation between CO and NO ($R^2$=0.49, p=0.01, Figure S6b)

suggests that CEC in the morning may have been influenced by emission from fresh complete combustion during the CEC case, whereas such correlations for PRD and CNC are not significant (Figure S6a and c). This indicates that the missing OH reactivity of CEC in the morning is possibly related to fresh vehicle emissions.

Diesel vehicle exhausts are rich in OVOCs relative to total VOCs (Yang et al., 2023). In our study, OVOCs were measured, except formaldehyde and acetaldehyde. We conducted a sensitivity analysis by adding these two OVOCs in the model (see Text S7 for details). After accounting for their influence, the overestimation of OH in the morning with constraining $HO_2$ could be significantly reduced (Figure 8a black line)."

Revised Text S7 on page 14 to 15 of the SI:

"A sensitivity test was conducted for the CEC case to account for the missing OH reactivity in the morning. This missing OH reactivity was attributed to unmeasured species in the fresh diesel exhaust. To estimate this, we first calculated the total OH reactivity of the exhaust based on the reactivity of $NO_x$ and CO, along with the diesel exhaust source profile. The contributions from $NO_x$ and CO were then subtracted. The remaining OH reactivity was allocated to formaldehyde and acetaldehyde, with their concentrations adjusted accordingly. This allocation was justified by the significant contribution of OVOCs to the total reactivity of diesel exhaust (Yao et al., 2015; Mo et al., 2016), as formaldehyde and acetaldehyde were not measured in this study. The sensitivity test was performed following these steps:

1. First, we calculated the OH reactivity of freshly emitted NOx and CO at each time step. We assumed that the pollutant concentrations at the time of the highest NO concentration did not undergo significant photochemical loss. For each time step, we calculated the ratio of the OH concentration at the time of the highest NO concentration to the OH concentration at that time step. This ratio was then multiplied by the OH reactivity of ambient $NO_x$ and CO at that time step to estimate the OH reactivity from the emitted $NO_x$ and CO.

2. The observed exhaust OH reactivity was determined by dividing the OH reactivity of emitted $NO_x$ and CO by 20%, which represents the minimum contribution of $NO_x$ and CO to the observed OH reactivity in exhaust in China (Yang, 2023).

3. The total exhaust OH reactivity was derived by dividing the observed exhaust OH reactivity by 60%, to account for the approximately 40% of OH reactivity missing in Chinese diesel exhaust (Yang, 2023). The OH reactivity of emitted $NO_x$ and CO was then subtracted from the total exhaust OH reactivity.

4. The remaining OH reactivity was allocated to formaldehyde and acetaldehyde in a 1:1.6 ratio, and their concentrations were adjusted accordingly. This ratio was calculated based on the concentration ratios of formaldehyde and acetaldehyde in diesel exhaust (Yao et al., 2015) and their respective OH reactivity coefficients."

**Q5:** *Figure 6: While the more significant discrepancies during the daytime should be the focus of the discussion, the model also underestimates HO₂ at night during CEC and CNC. This should also be mentioned in the manuscript and could be added to the discussion.*

**Response:** We speculate that the under-simulation of nighttime $HO_2$ is in part due to an underestimation of PAN concentrations by the model. However, other unknown processes must have a large contribution. We added a sensitivity test and discussion in section 3.3.3 on page 19 of the manuscript:

"**3.3.3 Nighttime underestimation of HO₂ in CEC case**

Ozone and $NO_3$ reactions with alkenes can produce $HO_2$ at night (Walker et al., 2015). In our study, alkenes are unlikely to be the main cause for the underestimation because the major alkenes were measured, and the alkenes concentrations in the CEC case were much lower compared to the PRD case in which no underestimation of nighttime $HO_2$ was found. A previous study (Whalley et al., 2010) showed that nighttime $HO_2$ underestimation at a clean tropical Atlantic site was significantly reduced by constraining the model with higher PAN. In our study, PAN was not measured. The model simulated nighttime PAN mixing ratios (0.1-0.7 ppb) were lower than previous observed nighttime results in the coastal (up to 1 ppb) (Xu et al., 2015) and mountain site (up to 2 ppb) (Wang et al., 2023) in southern China. To assess the impact of PAN concentration on nighttime $HO_2$ levels, a sensitivity analysis was conducted in which the PAN concentration was increased. The results show that only when the PAN concentrations were increased by tenfold, the model simulated nighttime PAN level could match the observations (Figure 8b, black line). This suggests that underestimated PAN might have contributed to the model's nighttime $HO_2$ underestimation, but other processes must have a larger contribution."

**Q6:** *Figure 3 and Section 3.1.2 – I suggest highlighting the different measurement periods in Figure 3 to better communicate which observations are included in the PRD, CEC, and CNC periods. At first glance, it appears that the majority of the highest observed HOₓ concentrations occurred during the three cases, and the lowest HOₓ concentrations (December 1-6 and 10-19) are omitted from the analysis. Were model runs also performed for these days?*

**Response:** Thank you for your suggestion. We have marked the different periods for the PRD, CEC, and CNC cases in Figure 3 in revised manuscript. On selection of CNC cases. We selected the three days with the highest and most consistent solar radiation for cases analysis. Other CNS cases with lower solar radiation (December 1-6 and 10-19) were not further analyzed.

We have added descriptions on the Section 3.1.2 on page 13 as follow:

"For each phase, a representative three-day period was selected for detailed analysis based on the availability of comprehensive data and sunny conditions (colored trajectories in Figure 4)."

**Q7:** *Figure S6: The standard deviation of AVOC and OVOC measurements increases suddenly during the daytime in the PRD case. Is it possible that the short gap in VOC measurements shown in Figure 3 is included in the average?*

**Response:** The increases of standard deviation of AVOCs and OVOCs in PRD are due to the missing data on afternoon of November 14th and the large variation between data collected on afternoon of November 12th and November 13th. Additionally, we decided to add Figure S6 to the main content (Figure5 in the new version) to provide diurnal variations information in the revised manuscript. We have added a detailed explanation of this occurrence to the Figure 5 captions and provide context to the observed data patterns on page 14:

"
[Figure]

[Figure]

**Figure 5** Average diurnal variations of (a) Temperature (b) Relative Humidity (c) $J_{O1D}$ (d) OH (e)HO$_2$ (f) Ozone (g) NO (h) NO$_2$ (i)HONO (j)Isoprene (k) AVOCs (l) OVOCs. The solid-colored lines represent selected cases: orange for PRD, green for CEC, and blue for CNC. The light band represents the standard deviations of the mean. The increase in the standard deviations of VOCs and OVOCs during the PRD case is a result of absence of data on the afternoon of November 14th and large variations in on November 12th and 13th."

*Minor comments:*

**Q8:** *The instrument to measure HONO is not listed in Table S2*

**Response:** We have updated Table S2 to include the instrument (LOPOP-03) used for HONO measurement on page 5 of the SI.

**Q9:** *Figure 5: The y-axis label should be ppb h$^{-1}$ not ppb s$^{-1}$ and OH + NO$_2$ should form HNO$_3$. In general, this figure is not easy to interpret due to the different axis scales and very small colored sections relative to reactions with NO. A total radical (RO$_x$) budget that does not include propagation channels may better illustrate the relative importance of initiation and termination processes in the model.*

**Response:** Thank you for pointing this out. The suggested revisions have been incorporated into Figure 6 (Figure 5 in previous version) on page 15 of the manuscript. Regarding the inclusion of a total radical ($RO_x$) budget, we note that due to the absence of direct measurements for $RO_2$, we lack a solid analytical basis to construct a reliable $RO_x$ budget. Therefore, we have not incorporated this metric, focusing instead on the OH and $HO_2$ we have measured and analyzed.

**Q10:** *Figure S4: As all data from the campaign is averaged together, this figure is misleading for some species that vary significantly from November to December such as isoprene or HONO. I suggest separating the data into two or three averaging periods or combining this figure with Figure S6 to illustrate how HONO, isoprene, and ozone changed during the transition from PRD to CEC and CNC.*

**Response:** In the revised Figure 5 (Figure S6 in previous version), we now present the average values for each of the three periods, addressing the significant variability of certain species like isoprene and HONO from November to December. This adjustment ensures a more accurate representation of how these species, along with ozone, vary during the transition from PRD to CEC and CNC.

**Q11:** *Tables 1, S2, and S4: Aligning text to the left of each column would improve readability. There is also a problem with the resolution of Table S2.*

**Response:** We have accepted the suggestion and improved the readability of Tables 1, S2, and S4 by aligning the text to the left in each column on page 11 to 12 of the manuscript, page 5 and page 7 of the SI, respectively.

---

## Author Comment (AC3)

**Response to Reviewers' Comments on Manuscript ID egusphere-2024-3210:**
**"Observation and modelling of atmospheric OH and HO₂ radicals at a subtropical rural site and implications for secondary pollutants"**

We sincerely appreciate the editor and the three referees for thoughtful comments and valuable suggestions which has helped us to improve our manuscript. In response to your comments, we have undertaken a comprehensive revision of the manuscript.

Please find our itemized responses below and revisions in the re-submitted files. We use *italicized* text for your questions, **blue** for our response, and **red** to indicate where changes have been made in the manuscript. The changes inside the manuscript will be highlighted in yellow.

*Reviewer 3*

*The manuscript reports about CIMS measurements of OH and HO2 in a subtropical rural site and the comparison of them with the results of MCM box model. Although in general simultaneous measurements of OH and HO2 provide very helpful information for understanding of radical budgets and the article reporting such measurements would potentially be of an interest, the present study cannot be published in its present form because it actually does not report HO2 measurements. The method used in this study for detection of peroxy radicals by their conversion to OH in reaction with added NO will result in about the same conversion efficiency for both HO2 and organic peroxy radicals RO2. To distinguish between the HO2 and RO2 radicals using the conversion scheme with NO several groups previously developed CIMS methods based on a modulation of chemical conditions in their instruments to measure either HO2 or RO2, predominantly (Hanke et al., 2002; Edwards et al., 2003; Hornbrook et al., 2011). In brief, HO2 mode requires efficient suppression of RO2 to HO2 conversion in reaction of alkoxy (RO) radicals with O2 in favor of the formation of alkyl nitrites:*

*RO2 + NO => RO + NO2*

*RO + O2 => R'O + HO2*

*RO + NO + M => RONO + M*

*Although the authors of the present manuscript make reference to the study of Edwards et al., 2003, they use NO concentration of 1.2 ppm leading to about 90% conversion of RO to HO2 and resulting in similar contribution of HO2 and RO2 to the detected signal, assuming their similar ambient concentrations. Referring in the manuscript supplement to the study of Fuchs, 2014 as an example of using the same NO concentration of 1.2 ppm for HO2 detection the authors do not take into account low pressure in the FAGE conversion stage, hence low O2, making RO+O2 reaction negligible and allowing HO2 measurements with low interference from RO2, although not for all of RO2 (Fuchs et al., 2011). Finally, the authors do mention once "interference" from RO2 by saying that "To access HO2 interference caused by the ambient RO2 conversion, the model underwent a three-days spins-up to simulate the ambient RO2 concentration". However, it doesn't seem to be a correct procedure to make correction using the model and after*

*that compare the measurements corrected in this way with the same model.*

*The present OH and "HO2" measurements may still be of value and present the basis of an important publication. However, for this the measurements should be correctly interpreted and presented with detailed description of a calibration procedure of the peroxy radicals.*

**Response:** Thank you for your insightful comment. In our study, both the experimental and modelling results did not show significant $RO_2$ interference under our environmental conditions. Therefore, we consider our measurements to be representative of ambient $HO_2$ concentration.

The details were added to the revised manuscript on page 7:

"Interference from $RO_2$ can affect $HO_2$ measurements, potentially resulting in an overestimation of ambient $HO_2$ levels (Edwards et al., 2003; Fuchs et al., 2014; Hanke et al., 2002). In our study, both the experimental and modelling results did not show significant interference under our environmental conditions (Text S4.3)."

The contents were added an Text S4.3 on page 11 on SI as below:

"To assess the possible $HO_2$ interference from $RO_2$, we first simulated with a box model production of $HO_2$ from $RO_2$ in the inlet system with addition of 1.2 ppm NO to ambient air composition observed in previous field study in Hok Tsui (a coastal site in Hong Kong) by our team in 2020 (Zou et al., 2023) before this observation. The observation-constrained MCM model (described in the Text S6) was run for three days, and the $RO_2$ outputs were taken as used as the initial concentrations entering the inlet. Then another model run was conducted by setting j-values setting to zero and reaction time as the residence time (47 ms) to simulate the conversion of $RO_2$ by NO in the CIMS inlet. We compared the OH concentrations (from $RO_2$ conversion to $HO_2$ and then to OH) at the outlet with the total concentration of $HO_2$+OH after spinning up. The result shows a difference of less than 2% suggesting negligible conversation of $RO_2$ to $HO_2$ in the inlet at 1.2 ppm NO injection. Similar model tests with real time conditions were also done for Conghua study after field study and show less than 2% interference.

To verity the model results, experiment tests were conducted in both laboratory and field settings (Hok Tsui in Hong Kong and Conghua) by comparing the $HO_2$ calibration factor obtained in synthetic air (with minimal interference of $RO_2$ due to very low VOCs concentrations in the synthetic air) and that conducted in indoor and outdoor air (with potential interference due to presence of VOCs). The results in Table S6 show difference of 1% - 3% between the $HO_2$ calibration factor in the synthetic air with that in the lab indoor air and that in the ambient air at Hok Tsui (with $[O_3]$ <70 ppb $[NO_x]$ < 10 ppb) as well as Cong Hua (with $[O_3]$ <60 ppb $[NO_x]$ < 10 ppb), confirming little interference of $RO_2$ to $HO_2$ measurements (See Table S6 below). These results might be due to the low concentrations of biogenic volatile organic compounds (BVOCs) in our two study sites ($[C_5H_8]$ <0.2 ppb) as previous studies show large interference of BVOC than anthropogenic VOC to $HO_2$ measurements (Fuchs et al., 2014).

**Table S5** Calibration factors of OH and HO$_2$ obtained in different conditions to estimate the RO$_2$ interference.

| Year of experiements | Calibration Conditions | Calibration Factor (cm$^3$) | | Notes |
|---|---|---|---|---|
| | | OH | HO$_2$ | |
| 2021 | Synthetic air in lab | 7.912E-10 | 9.156E-10 | On the afternoon of Nov 20, 2021 |
| | Indoor air in lab | 8.146E-10 | 9.275E-10 | |
| | Synthetic air at Hok Tsui | 8.212E-10 | 9.181E-10 | On the morning of Dec 23, 2021 |
| | Outdoor air at Hok Tsui | 8.252E-10 | 9.378E-10 | |
| 2022 | Synthetic air in lab | 1.043E-09 | 1.080E-09 | On the morning of May 04, 2022 |
| | Indoor air in lab | 1.035E-09 | 1.119E-09 | |
| | Synthetic air at Conghua | 1.033E-09 | 1.085E-09 | On the morning of Nov 17, 2022 |
| | Outdoor air at Conghua | 1.025E-09 | 1.092E-09 | |

Notes:

The difference between calibration factors obtained at 2022 and 2020 is due to the changes of CIMS's settings

Chemical Condition of outdoor air of Hok Tsui  [O$_3$] <70 ppb, [NO$_x$] < 10 ppb, [C$_5$H$_8$] < 0.2 ppb

Chemical Condition of outdoor air of Cong Hua  [O$_3$] <60 ppb, [NO$_x$] < 10 ppb, C$_5$H$_8$] < 0.2 ppb

"

As to the referee's comment on our correction of interference by comparing model and observations, the correction affected concentrations of less than 2%. We agree with the referee that this correction is take account for partial interference. However, the small correction is consistent with the test results shown above indicating minimal interference of RO$_2$ to HO$_2$ during the field study.

The following modification have been made on page 14 of SI as below:

"**Text S6 The model efforts to correct measurement interference.**

Ambient NO can cause interference to OH measurement by concerting HO$_2$ to OH in the inlet system. To assess and correct this effect, model simulations were conducted, which also simulated conversion of RO$_2$ to HO$_2$ by NO (i.e., interference to HO2 measurements as discussed before). We first constrained all measured species (except OH and HO$_2$) in the model, and a three-day spin-up was used to simulate the chemical conditions of the sampled air during measurement. Then the outputs were used as inputs for another simulation with the injection gases (SO$_2$ and/or NO) to simulate chemical reactions in the inlet with reaction time of 47 ms to match the reaction time in the CIMS. Photolysis frequencies were maintained at zero to reflect the dark environment of the inlet. The modeled OH concentrations without NO addition and OH concentrations with NO addition represent ambient NO interference to OH and HO$_2$, respectively.

The calculated interferences for OH and HO$_2$ measurements were in the range $1\times10^4$ cm$^{-3}$ to $1\times10^5$ cm$^{-3}$ (mean: $3\times10^4$) and $8\times10^5$ cm$^{-3}$ to $2\times10^6$ cm$^{-3}$ (mean: $1.2\times10^6$), respectively. These lead to correction of measurement data of OH and HO$_2$ less than 2%."

---

## Author Response (AR3)

We sincerely thank the reviewer for the thoughtful and detailed comments provided in this third round of review. We carefully considered each suggestion and revised the manuscript accordingly. In the response below, we use *italic font* to represent the reviewer's comments, **blue text** for our replies, and **red text** to quote content directly from the revised manuscript. Specific revisions are highlighted in **yellow** within the marked-up manuscript for clarity.

*I suggest to replace "HO2\*" by something similar to what was previously used to denote HO2+RO2 in numerous publications reporting peroxy radical measurements made with CIMS or PERCA, e.g. "sum of peroxy radicals". The "HO2\*" notation is not common, although it was used in the body of a few publications reporting HO2 measurements with FAGE.*

Thank you for the suggestion. Our instrument configuration during the field study did not measure all $RO_2$ radicals but a portion of them. So, the term "$HO_2 + RO_2$" or "sum of peroxy radicals are not suitable to our measurement. We would like to retain "$HO_2$\*" to represent the positively biased $HO_2$, which is analogous to the use of the term $NO_2$\* for positively biased $NO_2$ measured using a thermal catalytic conversion method. Using $HO_2$\* is also in line with its uses in recent publications.

*Abstract*

1. *p.1 l.19. "(HO2 + parts of RO2)" I think, better to say HO2 + contribution from RO2. Also define here the "RO2".*

    Thanks for the suggestions. We agree and have made the suggested revision as shown below:
    "This study measured $HO_2$\* ($HO_2$ + contribution from $RO_2$, organic peroxyl radicals) and OH concentrations……"

2. *p.1, l.22-23. "Model estimated interference to HO2 by RO2 possibly contributed to 44%-69% of the HO2\*".Is it rather estimated contribution from RO2 to HO2\* during the measurements period, ranging from 44% to 69%? Please reformulate.*

    To improve clarity, the sentence has been updated accordingly:
    "Model estimated contribution from $RO_2$ to $HO_2$ during the measurement period ranged from 44% to 69% of the $HO_2$\*."

*Introduction*

3. *p.2,l.9. "where R represents an alkyl group" There are many types of peroxy radicals, not just alkyl peroxy radicals!*

    Thanks for pointing this out. We have revised the text and the change is provided below:
    "……producing $HO_2$ and other organic peroxyl radicals ($RO_2$, where R represents an organic group such as alkyl, acyl, or aryl)."

4. *p.2,l.15. Define HOx*

    The definition of $HO_x$ was added as shown below:
    "$HO_X$ (OH + $HO_2$) radicals are removed……"

5. *p.3,l.26. Define "HO2\*" used here for the first time.*

    The definition of $HO_2$\* was added as shown below:
    "……concentrations of OH and $HO_2$\* ($HO_2$ + contribution from $RO_2$) using……"

*Methodology*

6. *p.4, l.25. Add information about HCHO to the Tables S2 and S4*

    Thank you. The measurement method and values for HCHO photolysis and other species (J

values) have been added to Tables S2 and S4 as shown below:

Table S2:

[revised manuscript text omitted]

7. *p.5,l.6. Replace "HO2" by "HO2+RO2" or "HO2*". The same for many other HO2 occurrences.*

We have replaced "HO$_2$" with "HO$_2$*" at this location and throughout the manuscript wherever it refers to the measured HO$_2$* values.

8. *p.7,l.18. "79% or 222%" Use the same way and numbers to present RO2 contribution (compare with given here and in the Abstract).*

We have changed the description of RO$_2$-contributed interference for consistency as follows:

"For our CIMS configuration, the model estimated daytime interference from RO$_2$ ranged from 44% to 69% of the HO$_2$* during the field study (Text S4.3 and Figure S9)."

9. *p.8, l.1-3.*

*1) Make the reported here calibration coefficients consistent with presented in Table S3;*

*2) For HO2 calibration it could be 46%, but for HO2\* it is up to 222% (see above)*

*3) How the background corresponding to H2SO4 mode was measured? As neither H2SO4 measurements nor H2SO4 calibration are presented here, the information about H2SO4 can be removed.*

We thank the reviewer for the suggestions. The reported calibration coefficients in the main text have been revised for consistency with Table S3. We also corrected the uncertainty of $HO_2^*$ to reflect model-estimated $RO_2$ interference, and removed the mention of $H_2SO_4$ background and calibration, as these data are not presented in the manuscript.

"The calibration factors, detection limits and uncertainties were $1.09 \times 10^{-8}$ cm$^{-3}$, $3 \times 10^5$ cm$^{-3}$, and 44% for OH; $1.07 \times 10^{-8}$ cm$^{-3}$, $2 \times 10^6$ cm$^{-3}$, and 222% for $HO_2^*$, respectively (Table S3). The large uncertainty in $HO_2^*$ reflects the possible contribution of $RO_2$ interference, as discussed above."

Table S3 was also revised as below:

| | | | | | a) Hok Tsui 2020 | | | | | | | | b) CongHua 2022 | | |
|---|---|---|---|---|---|---|---|---|---|---|---|---|---|---|---|
| Efficiency | Parameter | Gas | Values | Units | Specification for Measurement | Values | Units | Efficiency | Parameter | Gas | Values | Units | Specification for Measurement | Values | Units |
| $E_{Conv}$ | Front Injection | SO$_2$ (0.9%) | 5 | sccm | Sample Flow [SO$_2$] | 12 | ppm | $E_{conv}$ | Front Injection | SO$_2$ (0.9%) | 5 | sccm | Sample Flow [SO$_2$] | 12 | ppm |
| | | | | | | | | | | NO (0.9%) | 0.5 | | Sample Flow [NO] | 1.2 | ppm |
| | Pulse Valve | N$_2$ | 2 | sccm | Cycle Duration (OH) | 6 | mins | | Pulse Valve | N$_2$ | 2 | sccm | Cycle Duration (OH) | 6 | mins |
| | | | | | | | | | | | | | Cycle Duration (HO$_2$) | 60 | mins |
| | | C$_3$F$_6$ (99.9%) | 2 | sccm | B/S Ratio for OH measurement | | 8% | | | C$_3$F$_6$ (99.9%) | 2 | sccm | B/S Ratio (OH) | | 10% |
| | | | | | | | | | | | | | B/S Ratio (HO$_2$) | | 20% |
| | Rear Injection | C$_3$F$_6$ (99.9%) | 2 | sccm | Sample Flow [C$_3$F$_6$] | 1072 | ppm | | Rear Injection | C$_3$F$_6$ (99.9%) | 2 | sccm | Sample Flow [C$_3$F$_6$] | 1072 | ppm |
| | | HNO$_3$ | 10 | sccm | Reaction Time | 47 | ms | | | HNO$_3$ | 10 | sccm | Reaction Time | 47 | ms |
| | Sample Flow | | 3.7 | slpm | Sample Flow Speed | 55 | cm/s | | Sample Flow | | 3.7 | slpm | Sample Flow Speed | 55 | cm/s |
| $E_{Ion}$ | Sheath Flow | Zero Air | 12.6 | slpm | Reynolds Number in Ionization Chamber | >4000 Turbulent flows | | $E_{Ion}$ | Sheath Flow | Zero Air | 12.6 | slpm | Reynolds Number in Ionization Chamber | >4000 Turbulent flows | |
| | | HNO$_3$ | 10 | sccm | | | | | | HNO$_3$ | 10 | sccm | | | |
| | | C$_3$F$_6$ (99.9%) | 2 | sccm | Sheath Flow [C$_3$F$_6$] | 159 | ppm | | | C$_3$F$_6$ (99.9%) | 2 | sccm | Sheath Flow [C$_3$F$_6$] | 159 | ppm |
| | Total Flow | | 16.8 | slpm | Sheath Flow Speed | 25 | cm/s | | Total Flow | | 16.8 | slpm | Sheath Flow Speed | 25 | cm/s |
| | Sheath Voltages | | -80 | V | Voltages Difference for ionization | 48 | V | | Sheath Voltages | | -80 | V | Voltages Difference for ionization | 48 | V |
| | Sample Voltages | | -32 | V | | | | | Sample Voltages | | -32 | V | | | |
| $E_{Trans}$ | Buffer Ga | N$_2$ | 440 | sccm | Voltages Difference for transmission | 80 | V | $E_{Trans}$ | Buffer Ga | N$_2$ | 440 | sccm | Voltages Difference for transmission | 80 | V |
| | Buffer Voltages | | -70 | V | | | | | Buffer Voltages | | -70 | V | | | |
| | Pinhole Voltages | | -40 | V | | | | | Pinhole Voltages | | -40 | V | | | |
| Cal | Calibration Flow | | 10 | slpm | Calibration Factor C$_{OH}$ (Reagent ion: N$^{18}$O$_3^-$) | $1.21{*}10^{-8}$ | cm$^3$ | Cal | Calibration Flow | | 10 | slpm | Calibration Factors | C$_{OH}$ | $1.09{*}10^{-8}$ | cm$^3$ |
| | Flow Speed | | 65 | cm/s | | | | | Flow Speed | | 65 | cm/s | (N$^{18}$O$_3^-$) | C$_{HO2}$ | $1.07{*}10^{-8}$ | |
| | Product It Value | | $8.8{*}10^{10}$ photon/cm$^2$ | | | | | | Product It Value | | $8.8{*}10^{10}$ photon/cm$^2$ | | | | |
| Uncertainties | Sigma | | 2 | | Detection Limit ($\times10^5$ cm$^{-3}$) (3σ) | In lab | 1.7 | Overall Uncertainties (2σ) | | OH | 44% | | Detection Limit in Field Study ($\times10^5$ cm$^{-3}$) (3σ) | OH | 3 |
| | Calibration | | 38% | | | Day | 12 | | | HO$_2^*$ | 222% | | | HO$_2^*$ | 20 |
| | Overall | | 44% | | | Night | 8.5 | | | | | | | | |

10. *p.8,l.22-23 "Methacrolein (MACR), a derivative of isoprene, is distinctly classified among the biogenically sourced OVOCs for further discussion."*
   *I could not find any "further discussion".*

   The sentence mentioning methacrolein (MACR) has been removed from the updated manuscript, as MACR is neither discussed in the main text nor presented in Table 1 in the current version.

   *Results and Discussion*

11. *p.16,l.9. recycling is not a primary source*

   We have revised the sentence to clarify that recycling is not a primary source, but a dominant pathway under specific conditions. The updated text reads:

   "During midday (10:00–15:00), the recycling of RO species becomes the dominant pathway for $HO_2$ production, with rates of……"

12. *p.17,l.14-16 "The model calculated average daytime (08:00-16:00) RO2 interference increased HO2 by 127%, 117%, and 144% for PRD, CEC and CNC case, respectively." Reformulate with reference to Text 4S.3 to make it clear that it is about the estimated contribution of RO2 to HO2* signal? Also, see the comment to p.7,l.18 above.*

   We have revised the sentence to clarify that it refers to the model-estimated contribution of

$RO_2$ to the $HO_2^*$ signal. The updated sentence now reads:

"According to model simulations (Text S4.3), $RO_2$ interference was estimated to account for 56%, 54%, and 59% of the observed $HO_2^*$ signal for the PRD, CEC and CNC case, respectively."

Revised content on Text S4.3

…, we determine that the average daytime (08:00-16:00) $RO_2$ interference was estimated to contribute 56%, 54%, and 59% of the $HO_2^*$ signal during the PRD, CEC and CNC case, respectively. Throughout the entire campaign, the contribution ranged from 44% to 69%.

13. *p.18,l.9 Figure 8. Replace "PRD" by "CEC"*

Thanks for comment, the figure notion has been corrected by replacing "PRD" with "CEC" as suggested.

14. *p.19,l.9 "substantially higher modeled HO2 concentration than base model" Or than HO2*(obs)?*

Thank you for the suggestion. We have revised the sentence to correctly compare the modeled $HO_2$ with the observed $HO_2^*$ as follows:

"……, while constraining OH still leads substantially higher modeled $HO_2$ concentration (blue line in Figure 7b) than the observed $HO_2^*$."

15. *p.19,l.18 "suggesting that there may be missing OH reactivity" It looks more like the result of constraining the model with high HO2 in combination with early morning NO peaks. Hence, either erroneously estimated HO2, or, as suggested, some missing OH loss.*

Thank you for the insightful comment. The reviewer is correct that the early morning OH overestimation could result from either the overestimated $HO_2$ constraint or from missing OH loss processes. In our model setup, the constrained $HO_2$ values are already close to the observed $HO_2^*$, representing the potential upper limit of ambient $HO_2$. Nonetheless, we agree that the possibility of erroneous $HO_2$ input cannot be excluded. Therefore, we have revised the sentence to reflect both potential explanations.

"However, the OH concentration is overestimated in the morning when the corrected $HO_2^*$ was constrained, suggesting that some OH sinks may be missing in the model during this period or the corrected $HO_2^*$ values that were used to constrain the model are still higher than the true $HO_2$ values."